# Cleaning Costs for European Sheltered White Painted Steel and Modern Glass Surfaces Due to Air Pollution Since the Year 2000

**Terje Grøntoft [1,\*]**, **Aurélie Verney-Carron [2]** and **Johan Tidblad [3]**

[1] NILU-Norwegian institute for air research, Instituttveien 18, Box 100, NO-2027 Kjeller, Norway
[2] LISA-Laboratoire Interuniversitaire des Systèmes Atmosphériques, UMR CNRS 7583, Université Paris-Est Créteil, Université Paris-Diderot, Institut Pierre-Simon Laplace, 61, avenue du Général de Gaulle, 94010 Créteil, France; Aurelie.Verney@lisa.u-pec.fr
[3] Swerea KIMAB, Isafjordsgatan 28A, 164 40 Kista, Sweden; johan.tidblad@swerea.se
\* Correspondence: teg@nilu.no; Tel.: +47-6389-8023

**Abstract:** This paper reports estimated maintenance-cleaning costs, cost savings and cleaning interval increases for structural surfaces and windows in Europe obtainable by reducing the air pollution. Methodology and data from the ICP-materials project were used. The average present (2018) cleaning costs for sheltered white painted steel surfaces and modern glass due to air pollution over background, was estimated to be ~2.5 Euro/m$^2$·year. Hypothetical 50% reduction in the air pollution was found to give savings in these cleaning costs of ~1.5 Euro/m$^2$·year. Observed reduction in the air pollution, from 2002–2005 until 2011–2014, have probably increased the cleaning interval for white painted steel with ~100% (from 12 to 24 years), representing reductions in the single intervention cleaning costs from 7 to 4%/year (= % of one cleaning investment, per year during the cleaning interval) and for the modern glass with ~65% (from 0.85 to 1.3 years), representing reductions in the cleaning cost from 124 to 95%/year. The cleaning cost reductions, obtainable by 50% reduction in air pollution, would have been ~3 %/year for white painted steel and ~60%/year for the modern glass, representing ~100 and 50% additional cleaning interval increases. These potential cleaning cost savings are significantly higher than previously reported for the weathering of Portland limestone ornament and zinc monuments.

**Keywords:** soiling; modern glass; facades; air pollution; maintenance costs; cleaning costs; cleaning interval; dose-response function

## 1. Introduction

Air pollution is damaging health, ecosystems and built structures [1–4], it represents large costs to society and weakens its sustainability [5]. Management of the air pollution to improve air quality has large benefits in reducing such damages and costs. Monitoring of the air quality is being performed locally, regionally and in international networks for research and regulation purposes [6,7]. The Convention on Long-range Transboundary Air Pollution [8] is one such network, which has since 1979 addressed some major environmental problems related to air pollution, through scientific collaboration and policy negotiation. Within its Working Group on Effects the convention include the ICP-materials programme (the International Co-operative Programme on Effects on Materials including Historic and Cultural Monuments, within the Convention on Long-range Transboundary Air Pollution (CLRTAP), organized under the United Nations Economic Commission for Europe (UNECE)) [9], which has since 1985 worked to assess the effects and trends of corrosion and soiling of materials caused by air pollution. Atmospheric weathering, corrosion and soiling degrade outdoor

built surfaces and are major reasons for maintenance work. The rate of the degradation depends on the atmospheric conditions, including the amount of air pollution, which will affect the frequency and cost for the maintenance [10]. A significant part of this cost would be due to air pollution from local pollution drivers, above the background levels representing mainly transboundary transports of polluted air [11,12]. This emphasises the anthropogenic factors driving the cost development in this field, and the possibility for mitigation policies to further reduce the air pollution and related maintenance costs.

During the second part of the 20$^{th}$ century, negative impacts from high $SO_2$ concentrations in air, coming from the burning of fossil fuels, was a major concern for health, ecology and built structures in Europe. Control of the $SO_2$ emissions and changes in the industrial sector were implemented to address this problem. This resulted in large reductions in the $SO_2$ concentrations [13,14]. Besides the beneficial effects for health and ecology, this significantly reduced the atmospheric chemical weathering and corrosion of built structures [15]. In the new multi-pollutant situation, several other air pollutants in addition to $SO_2$ significantly influenced the atmospheric corrosion of materials [16–18]. In this situation, the focus for most ongoing environmental measurements, which were performed to assess health risk and the meeting of air quality standards, were shifted away from $SO_2$ to continue for nitrous gases (NOx) and airborne particulate matter (PM). Additionally, the soiling of outdoor surfaces, and related maintenance and cleaning costs, was becoming increasingly more important in comparison with corrosion effects and costs. This led to the initiation in the ICP-materials programme and the connected EU project Multi-Assess of soiling measurements on a range of material surfaces including modern glass, in parallel with ongoing material corrosion measurements. Simultaneous measurements were performed of the main influencing environmental parameters [17,18]. As huge areas of building facades, windows and other surfaces of built structures and monuments are gradually weathered and become soiled, the total cost of their maintenance or cleaning is clearly large.

Soiling is the visual effect resulting from the darkening of exposed surfaces following the deposition and accumulation of atmospheric particles (Figure 1, [19–22]). Soiling is a complex process depending on factors related to the particles, the gaseous pollutants, the surfaces, the local meteorology and rainwater. Soiling is often not a separate process from the weathering and corrosion of surfaces. Biological processes, such as the growth of bacteria, algae and fungi, are often important. Darkening by soiling and corrosion can happen due to deposition both of particulate and gaseous pollutants. A good example is the common formation of black crusts on calcareous stone. This is a result of deposition of $SO_2$, which leads to the formation of gypsum, and of discolouring particulate matter (soot) [10,23]. Soiling can be assessed by the change of optical properties of different materials, generally, as the loss of reflectance for opaque materials (such as stones, concrete, painted steel, etc.), and as the loss of transparency, i.e. transmittance, for transparent materials (such as glass).

Air pollution dose-response functions (DRFs; the term "exposure response function" (ERF) is also used, for example in Reference [24] when discussing costs, however, this work will apply the term "dose response function", which is usually used to describe the corrosion-weathering functions, and we will use this term throughout the work) for the soiling were developed connected to the work in the ICP-materials programme and the EU Multi-Assess project [10,13,25–29], as had earlier been developed for atmospheric corrosion of materials [20,21,30]. Most of the measurements were carried out on samples that were exposed in vertical position under sheltered conditions, as rainwater would induce recurring washing (see Figure 2 for an example of soiling measurements on modern silica-soda-lime glass performed on a station in the ICP-materials programme). Therefore, only the sheltered condition will be considered in this paper.

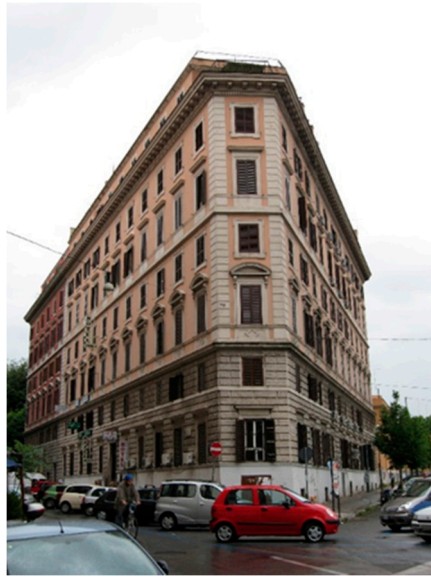

**Figure 1.** Soiling of a building facade and windows in Rome, Italy.

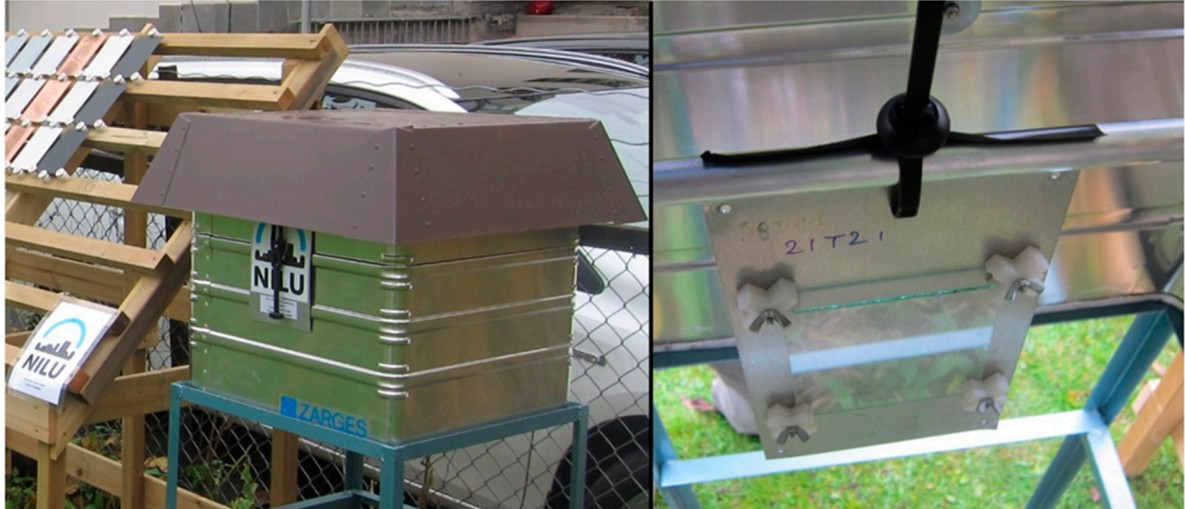

**Figure 2.** Soiling measurements on modern glass performed on the ICP-materials station in Oslo (2017) (The Oslo station is not included in Tables A1–A3, as particulate matter concentration was not measured at the station).

The dose-response functions have been combined with methods for the evaluation of corrosion costs and cost savings associated with pollution reductions [31]. However, lack of consistency in the way scenarios have been constructed and compared have resulted in differences in reported cost estimates, and the estimates have been difficult to generalise to other scenarios for future pollution reductions. Later than and beyond the references provided in Reference [31] and in this paper, major reporting of estimates for cleaning costs of built surfaces due to air pollution in Europe, has, to our knowledge, not been published.

The goal of this paper was to apply measurement results from the ICP-materials programme to estimate cleaning (soiling) costs for buildings caused by the air pollution in Europe. Recent trends (since the year 2002) in cleaning costs, and possible savings due to 50% reduction in air pollution, were estimated for one opaque material: white painted steel, as representing white painted smooth building facades and structural surfaces; and one transparent material: modern glass, in a position sheltered from precipitation. The estimations applied the method developed through the ICP-materials and related projects since the 1980s for the Europe wide network of ICP-materials

locations. This "bottom-up" approach estimates costs from the expected physical soiling progress, rather than from renovation cost records (so-called "top-down estimations [24]). A comparison was then made with previously reported costs due to atmospheric chemical weathering (corrosion) of zinc and Portland limestone surfaces exposed outdoor in Europe since the 1980s, and of savings due to a similar amount of hypothetical pollution reduction (50%) [11].

It is a major aim in air quality management to reduce the air pollution and related negative impacts and costs, to meet, and if possible obtain cleaner air than, agreed targets. A 50% reduction in the air pollution is often a realistic aim. Calculation of cost savings obtainable by a 50% reduction in air pollution and comparison with targets and guidelines, illustrate the effect of reducing the air pollution well. Estimations for single locations would be directly relevant for policies to address the air pollution impacts at the locations.

Building surface renovation is a huge continuous effort. The soiling is one of many reasons for renovation. It can be difficult to determine the interaction between and relative influence of different deterioration processes on the overall condition of a building surface. It seems useful to distinguish the cleaning due to soiling of facades from the multitude of other maintenance actions. The soiling can be a main reason for cleaning, for example of windows. However, the soiling should be evaluated in the context of other deterioration processes, and other maintenance actions than cleaning. This study contributes with values for the soiling costs in Europe. Their relevance should be evaluated together with other deterioration impacts on built structures and costs. The results do not directly represent situations outside of the ranges of the input data, for example outside of Europe, in different climates or different air pollution situations. It should be further noted that the calculated costs reported in this work do not include amenity loss or discounting (see Discussion chapter).

## 2. Methods

### 2.1. The Dose-Response Functions

This paper deals with the soiling of an opaque material, white painted steel [32], and a transparent material, modern silica-soda-lime glass (Si-Na-Ca float glass, Planilux®) [25]. The following dose-response function (Equation (1)) for the accumulated soiling of a white painted steel surface [10,32] was used to indicate the development of outdoor surface soiling from measured environmental conditions, to allow comparison and averaging over the European area. It is likely to quite well represent white painted smooth facades and surfaces, which are common. It may less well represent perceived soiling and cleaning cost for other kinds of surfaces (see Discussion):

$$\Delta R = R_0 \cdot \left( 1 - \exp\left( -[PM_{10}] \cdot t \cdot 3.96 \cdot 10^{-6} \right) \right) \tag{1}$$

where $\Delta R$ is the loss of reflectance (%) relative to the reflectance of the non-soiled surface, $R_0$ (%) at time zero, $[PM_{10}]$ is the concentration of $PM_{10}$ (particulate matter $\leq$ 10 μm in aerodynamic diameter, μg/m$^3$) and t the time of exposure (days). The constant (=3.96 × 10$^{-6}$) then has the unit m$^3$/μg·day.

Equation (1) was developed by non-linear regression after one year of experimental data measured in Athens, Krakow and London, of soiling, and $PM_{10}$ ranging from 20 to 80 μg/m$^3$ [32], in the EU Multi-Assess project [18]. This is in the upper range of the $PM_{10}$ values measured at the ICP-materials stations (see Data section). It should be used with more caution at $PM_{10}$ values below about 20 μg/m$^3$, which was the case for 18 out of 19 data points (annual averages) for the rural stations and seven out of 28 data points for the urban stations, in the ICP-materials data base from year 2002 to 2014. It has a physical basis, but it should still be considered that particle deposition is a complex phenomenon and that such relatively simple functions do not always best fit the experimental data [10].

The following two dose-response functions for the formation of haze on modern glass were used. Haze (H) is defined as the ratio between the diffuse transmitted light and the direct transmitted

light [33] expressed in percentage. One function is a statistical model equation, called NEUROPT-Glass, reported by [25]:

$$H_{est} = 4.81 \cdot H_{norm} + 5.27$$
$$H_{norm} = 3.951 - 39.193 \cdot \tanh(S_1) + 44.967 \cdot \tanh(S_2)$$
$$S_1 = -1.498 - 0.145 \cdot \left(\frac{t-387.18}{257.17}\right) + 0.031 \cdot \left(\frac{[SO_2]-9.7}{11.82}\right) + 0.297 \cdot \left(\frac{[NO_2]-33.29}{19.37}\right) + 0.28 \cdot \left(\frac{[PM_{10}]-28.93}{15.68}\right) \quad (2)$$
$$S_2 = -1.45 - 0.073 \cdot \left(\frac{t-387.18}{257.17}\right) + 0.033 \cdot \left(\frac{[SO_2]-9.7}{11.82}\right) + 0.281 \cdot \left(\frac{[NO_2]-33.29}{19.37}\right) + 0.261 \cdot \left(\frac{[PM_{10}]-28.93}{15.68}\right),$$

where $H_{est}$ is the estimated haze (%), and $H_{norm}$ is the normalized haze (%). The other function is a multilinear regression equation reported by [27]:

$$H = (0.2529 \cdot [SO_2] + 0.108 \cdot [NO_2] + 0.1473 \cdot [PM_{10}]) \cdot \left(\frac{1}{1 + \left(\frac{382}{t}\right)^{1.86}}\right), \quad (3)$$

In both Equations (2) and (3), t is the time (days). $[SO_2]$, $[NO_2]$, $[PM_{10}]$ are the concentrations in air ($\mu g/m^3$) of sulfur dioxide, nitrogen dioxide, and particulate matter with average aerodynamic diameter less than 10 μm.

Equation (2) is a pure statistical model developed by non-linear parametric regression, with the use of a hyperbolic tangential function in order to predict the evolution of haze. Its development was based on a neural network approach [25,28]. It should be used for times less than 1638 days and $[SO_2]$, $[NO_2]$, $[PM_{10}]$ less than 51, 90 and 84 $\mu g/m^3$, respectively. In the present study, the maximum time was set to 1500 days, equal to the cleaning interval in the background (see Section 2.4) and the concentration criteria were fulfilled (see Data section), The development of Equation (3) was based on multiple linear regression. The haze is a function of a temporal trend (Hill's equation), of which the amplitude is controlled by $SO_2$, $NO_2$ and $PM_{10}$ concentrations in the atmosphere [27]. The whole set of ICP-materials soiling-haze and environmental data until 2011 was used to test and validate both of the dose-response functions (Equations (2) and (3)) [28,34]. For the ICP-materials stations, the environmental parameter values for the "present" situation (the measurement years) are known. When the loss of reflectance ΔR or the haze H before cleaning are set, then the corresponding lifetimes, t, until the cleaning action can be analytically calculated from Equation (1) and numerically for Equations (2)–(3). The present average expected cleaning frequency was calculated for the ICP-materials locations, for each year of the environmental measurements, assuming constant future environments as in the measurement years. The cleaning frequencies for the hypothetical 50% reduced pollution situations were calculated similarly.

*2.2. The Calculation of the Costs*

A "standard method" was used here to calculate the costs for maintenance-cleaning of the white painted steel surfaces and modern glass due to air pollution over background levels [10,31,35]:

$$K_{p,r} = M \cdot P \cdot (1/t_{p,r} - 1/t_b), \quad (4)$$

where $K_p$ and $K_r$ (Euro/year) are cleaning costs due to air pollution in the present (p) or a reduced (r) air pollution situation, M ($m^2$) is the area of stock at risk for soiling of a type of material, P (Euro/$m^2$) is the single investment cleaning costs per meter squared of the material, and $t_{p,r}$ and $t_b$ (days or years) are the "lifetimes" between cleaning, i.e., the cleaning frequency or interval, in the present (p) or some reduced (r), and the background (b) pollution situations.

The saving in cleaning costs due to reduction in air pollution (ΔC in Euro/year, or in % of one cleaning investment, per year during the cleaning interval, when a value of 100 Euro for the cost M·P is used) is given by:

$$\Delta C = K_p - K_r, \quad (5)$$

The cleaning cost saving is the total saving, which does not depend on the background (the background terms disappear by subtraction in Equation (5)). The cleaning cost saving over several years could be calculated by multiplying $\Delta C$ (Euro/year) with the number of years. It is practical to apply the unit of % of the initial investment, rather than reporting the % saving of the yearly cleaning cost, as values for actual cleaning investments are more often known and more easily available than yearly costs (see also Discussion). Thus, by the terms in Equation (5), the reporting of results in this work is as $100 \times (K_p - K_r)/\Delta C$ (% of one cleaning investment, per year), rather than $100 \times (K_p - K_r)/K_p$ (% of the yearly cleaning cost). The unit used is below sometimes reported as (%/year), for example with results values in the text, to avoid tedious repetitions of the full long unit definition.

### 2.3. Data

The calculations were made for a selection of 12 urban, three industrial and eight rural sites of the ICP-materials program (see locations on Figure 3 and calculated values for the sites in Appendix A). The results reported in the Appendix tables represent the data availability at the ICP-materials stations [28]. The classifications of stations as urban (U), industrial (I) or rural (R) are also reported in the tables.

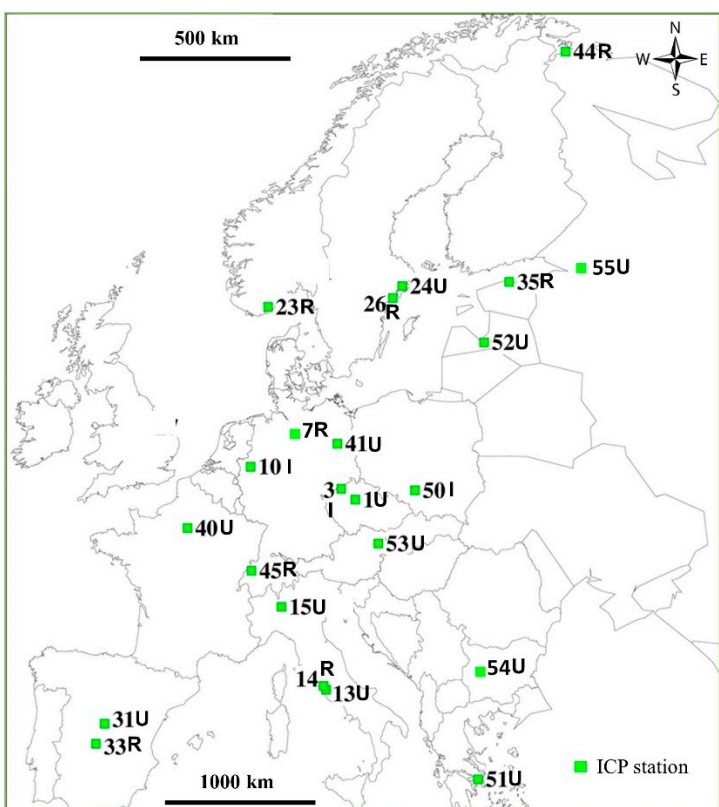

**Figure 3.** Map of the ICP-materials exposure locations. I = industrial, U = urban, R = Rural. Station names and data points are given in Tables A1–A3.

Environmental data from the ICP-materials stations from 2002 to 2011 [28] and 2014 [36] were used to calculate the time series estimates of facade/surface and window cleaning costs and savings (changes) resulting from reduction in air pollution, and of the cleaning intervals, for the average of the ICP-materials stations and the sub-selections of industrial (I), urban (U) and rural (R) stations, by Equations (1) to (5). The number of measurement stations were for each year: 2002: 4, 2005: 9, 2008: 15, 2011: 14 and 2014: 18, with fewer entries for each of the industrial (I), urban (U) and rural (R) sub-selections (Appendix A). The pollution data from the ICP stations are reported to the ICP-materials data base as monthly average values obtained with different measurements techniques (continuous

monitoring or passive sampling) with different initial resolution [9]. The minimum, 10th percentile, average, 90th percentile and maximum concentration values for the annual station data (in μg/m³), from year 2002 to 2014, were for $SO_2$ (0.1, 0.5, 5, 14, 39), for $NO_2$ (0.6, 1.9, 23, 46, 67) and for $PM_{10}$ (3.8, 7, 24, 48, 62). The trends for the pollution values measured at the ICP-materials stations since 1987 were strongly negative for $SO_2$ and clearly negative for $NO_2$ until about year 2000. From the year 2002, a continuing weak negative trend was observed for $SO_2$. For $NO_2$, and for $PM_{10}$ which was measured from year 2002, possible weak negative trends were observed at the urban and rural stations, but no clear trends at the industrial stations, as can also generally be seen in Table 1. The pollution trends are reported in References [15,28]. There were changes in the stations, and years with measurements for the stations, over the years from 2002 to 2014. Therefore, the calculations represent levels of the cleaning costs and savings, but only indications of trends over the years.

Table 1 gives the average values for the environmental data measured in 2002 (2005 for $PM_{10}$) and 2014, for all the stations included with measurements in the ICP-materials programme, and in 2005 and 2014 for the sub-selections of industrial (I), urban (U) and rural (R) stations. To show both the levels and trends, in Table 1, only the stations where measurements were performed in both year 2002 (or 2005) and 2014 (differently from the overall assessment below where all the available data for all the stations are included, also when a station did not measure in year 2002 (2005 for $PM_{10}$) and/or 2014 [28,37].), and in most cases in the years in between, are included in the averages. The table also shows the mean values measured at the ICP-materials station in Oslo, Norway, from 2002 to 2014 and used as input to the mapping example provided for Oslo. In accordance with the ICP-materials reporting, the data represent annual mean values for the measurements, which started in October in the given years.

**Table 1.** Environmental values used for the calculation of cleaning costs and savings for the sheltered smooth white painted facades/surfaces and modern glass. Annual average values for all the ICP-materials stations and for the sub-selections of industrial (I), Urban (U) and Rural (R) stations, with measurement in both 2002 or 2005, and 2014, and mean ICP station values for the years 2002–2014 and the input values to mapping for Oslo. n.a. = not available. Values in brackets are the number of measurement stations included. "-" = irrelevant.

| Para Meter | ICP-Mean 2002 ($PM_{10}$: 2005) | ICP-Mean 2014 | ICP-Industrial 2005 | ICP-Industrial 2014 | ICP-Urban 2005 | ICP-Urban 2014 | ICP-Rural 2005 | ICP-Rural 2014 | ICP-Oslo (Mean 2002-14) | Mapping Values: Oslo ICP Location, Mean Oslo |
|---|---|---|---|---|---|---|---|---|---|---|
| $SO_2$ | 7.0 | 3.5 | 23.5 | 11.2 | 4.6 | 2.4 | 2.0 | 1.6 | 2.1 | 3, 3 |
| (μg/m³) | (20) | (20) | (3) | (3) | (8) | (8) | (7) | (7) | (1) | (1), - |
| $NO_2$ | 23.2 | 20.5 | 33.8 | 35.5 | 30.2 | 25.8 | 3.6 | 3.3 | 28 | 28, 14 |
| (μg/m³) | (19) | (19) | (3) | (3) | (8) | (8) | (7) | (7) | (1) | (1), - |
| $PM_{10}$ | 25.6 | 20.4 | 36,5 | 30.4 | 37.6 | 30.2 | 9.9 | 5.7 | n.a. | 21, 10 |
| (μg/m³) | (7) | (7) | (3) | (3) | (2) | (2) | (2) | (2) | | (1), - |

The values of measured haze on modern glass for the ICP-materials stations are available in References [28,37].

## 2.4. Tolerable Soiling, Background and Comparison with Air Quality Guidelines

The cleaning frequencies for the tolerable level and background were calculated on the basis of the methodology previously used for corrosion in ICP-materials [27] and the MultiAssess [29] projects. This methodology suggested the background situation to correspond to the 10th percentile of measured values for the ICP-materials measurement stations, and the tolerable maintenance intervals for a future (year 2050) target to be two times the background value [38]. In this work the background values were calculated based on the 10th percentile criterion. The evaluation of the tolerable levels were made to best represent the typical cleaning interval, and thus costs, rather than fixing them to two times the background value. It was assessed that the average calculated values for the cleaning intervals for the ICP-materials stations, according to Equations (1)–(3), would best represent the typical cleaning intervals and costs.

For the white painted steel, the tolerable (the term "acceptable" is according to Reference [16] reserved for materials used in technical constructions while "tolerable" is used in connection with degradation of cultural heritage(; the term "tolerable" is used in this paper, covering both meanings) soiling before cleaning was set to 35% loss of reflectance ($\Delta R$) relative to the non-soiled surface ($R_0$) as has been found to trigger significant adverse public reaction [10,24]. The cleaning interval in the background was evaluated as the 90th percentile, and the cleaning interval in a tolerable pollution situation as the average of the calculated cleaning intervals until tolerable soiling, according to Equation (1), for all the ICP-materials stations and years when $PM_{10}$ was measured from 2002 to 2014 (Table A1, [28,36]). This gave a cleaning interval for the background of 34.9 years, and for the tolerable pollution situation of 18.2 years. For white painted steel, negative costs are reported in a few instances when the pollution values were lower than representing the background value, by Equation (1).

For the modern glass, there are (to our knowledge) no reported suggestions for the tolerable amount of haze before recommended cleaning. It was evaluated here that an annual cleaning interval would represent a typical practice and most relevant average. This corresponds to an average value measured for all the ICP-materials measurement stations and years, of 3% haze [28,37]. Haze is used in the glass industry to measure transparency and to express the idea of a visual nuisance that human eyes perceive when looking through a glass plate. A value of 1% corresponds to the perception of a "dirty" window [27] (picture examples are given in Reference [39]). With detection of haze when it approaches 1%, it seems realistic that a value of about 3% will trigger cleaning action. The level of tolerable haze before cleaning would, however clearly vary for different situations (housing, shops, office buildings and cultural heritage buildings) and branding requirements.

The annual background value for the haze was found to be 1.3%, representing the 10th percentile of the measured annual haze values for the ICP-materials stations from 2005 to 2014. The calculated annual station values, which most closely approximated the background value of 1.3%, were for Equation (2) rural station no. 23, Birkenes in 2014, with a haze value of 1.86, and for Equation (3) rural station no. 14, Casaccia in 2014, with a haze value of 1.36 [28,37]. Due to low measured pollution values the assessed maximum tolerable haze of 3% could not be reached, at long cleaning interval, by the calculation with either Equation (2) (for which time has to be less than 1638 days) or Equation (3) (which tends toward a plateau). The maximum haze that could be calculated for Birkenes (Equation (2)) was 2.1% at a lifetime of ~1500 days (~4 years), and for Casaccia (Equation (3)), it was 2.8% at a hypothetical lifetime of ~10,000 days (~27 years). These values are uncertain for longer times of haze development and it is common practice to clean windows more frequently than facades. Therefore, it was decided to use 1500 days (~4 years) as a maximum cleaning interval for situations where longer hypothetical intervals were calculated by Equations (2) and (3). Thus, the cleaning interval in the background was set to 1500 days in the calculations with both equations.

It is important to compare these calculated background and tolerable cleaning intervals with the health related guidelines and directives, which mostly guide policy (ICP-materials should, as a working group on effects within the Convention on Long-range Transboundary Air Pollution (CLRTAP), relate to the discussions and interactions about general air pollution policies considered to be important by the Convention), such as the most recent WHO (World Health Organization) guideline from the year 2005 [40] and the EU 2008 Air Quality Directive (AQD) presently in force [41]. The WHO guideline and EU 2008 AQD determine annual average concentrations for $PM_{10} = 20$ $\mu g/m^3$ (WHO) and 40 $\mu g/m^3$ (EU 2008 AQD), and $NO_2 = 40$ $\mu g/m^3$ for protection of health. Annual mean health limits are not reported for $SO_2$. Therefore, an $SO_2$ value of 20 $\mu g/m^3$ for vegetation, given in the EU 2008 AQD, was applied in both cases.

The cleaning intervals for white painted steel, representing the WHO guidelines and the EU 2008 AQD, were calculated by inputting the guideline values in Equation (1). The respective cleaning intervals for modern glass were set equal to the calculated cleaning intervals, by Equations (2)–(3), for the ICP-materials stations and years with similar first year calculated haze as from the pollution values in the WHO guidelines and EU 2008 AQD, by the following procedure:

The first year haze calculated from the pollution values in the WHO guidelines was: by Equation (2) = 4.9% and by Equation (3) = 5.9% haze, and calculated from the pollution values in the EU 2008 AQD it was; by Equation (2) = 6.8% and by Equation (3) = 7.3% haze. The stations, years, first year calculated haze and cleaning intervals before a tolerable haze of 3%, representing the WHO guidelines, were for: Equation (2), station 52, Riga in 2008 (4.9%, 163 days), and Equation (3), station 10, Bottrop in 2005 (5.9%, 209 days) and representing the EU 2008 AQD; Equation (2), station 51, Athens in 2011 (6.8%, 142 days), and Equation (3), Athens in 2008 (7.5%, 175 days). For the EU 2008 AQD and Equation (3), the calculated cleaning interval was adjusted according to the difference in the first year haze (in Athens 2008) from the first year value representing the EU 2008 AQD as follows. The expected haze value representing the EU 2008 AQD was 7.3% as compared to 7.5% in Athens, 2008. Due to the slightly lower air pollution and haze values representing the EU 2008 AQD than for Athens in 2008, the cleaning interval representing the EU 2008 AQD would be slightly longer. Therefore, the calculated tolerable lifetime in Athens in 2008, calculated to be 175 days, was adjusted by multiplying with 7.5%/7.3% to obtain a cleaning interval of 180 days representing the EU 2008 AQD.

Table 2 gives the cleaning intervals in the background, for the assessed tolerable level and representing the health guidelines, for the white painted steel (Equation (1)), and for the modern glass as the averages of the results from Equations (2) and (3).

**Table 2.** Cleaning intervals in the background, for the assessed tolerable level and representing health guidelines.

| | Soiling of White Painted Steel | Haze on Modern Glass |
|---|---|---|
| | Cleaning Interval (Years) | |
| Background value | 35 | 4.1 |
| Tolerable level | 18 | 1.0 |
| WHO (World Health Organization) guidelines | 15 | 0.5 |
| EU 2008 Air Quality Directive | 7.5 | 0.4 |

A comparison was made with the tolerable level for the averages of all the ICP-materials sites and years. For both the white painted steel and modern glass, it can be seen from Table 2 that the tolerable level is stricter than the WHO guidelines and EU 2008 AQD, as it recommends longer cleaning intervals.

Costs for the same cleaning operations are expected to vary significantly depending on the location, type of constructions and surfaces, economic, technological and other factors. Therefore, the "cleaning cost due to air pollution over background" were initially presented in the more general unit of: % of one cleaning investment, per year during the cleaning interval. This is the part of (percentage relative to) the cost for one cleaning operation, which would be spent every year to clean soiling from air pollution over the background level. Thus, the basis for the comparison of costs at different locations in different years is similar absolute costs, not similar cleaning areas. The areas, which could be cleaned at that cost, would be different between locations and, generally, have become less over the years. It would have been very difficult to obtain correct and comparable absolute costs for all the station locations and years included in the evaluation.

To recalculate to the unit Euro/year, any reported value (of % of one cleaning investment, per year during the cleaning interval) could be multiplied with M·P/100. An evaluation of the absolute costs and savings in Euro is given in the Discussion chapter. The presented results should be considered as indications of the cleaning costs and possible costs savings obtainable by reducing air pollution. Variations in cleaning costs and savings are expected depending on properties both related to the soiling and to other factors, as will be discussed in the Discussion chapter.

*2.5. Mapping*

The mapping example, which is presented for the main central Oslo urban area, used gridded pollution and climate values, with a resolution of one times one km, over an area of 22 km (east-west) times 18 km (south–north). Gridded values for the air pollutants, $NO_2$ and $PM_{10}$, were obtained from emission-dispersion modelling for the year 2003, with modelled values adjusted to measured values for the pollution species at central urban background and traffic stations [42,43]. A constant value for the concentration of $SO_2$ was used for the whole grid. Reports about air quality issued by the municipality of Oslo [44,45], and data from the ICP-materials station in Oslo [28], show no significant trends since 2002 in the concentrations of air pollutants causing soiling in Oslo (Equations (1) to (3)), except some possible decrease in $PM_{10}$ after 2005. The averages and ranges of the mapping input variables were: Variable (Mean, Minimum, Maximum) = $NO_2$ (14, 4, 50 µg/m$^3$), $SO_2$ (3 µg/m$^3$), $PM_{10}$ (10, 5, 30 µg/m$^3$). The mean mapping values over the grid and the mean values measured at the Oslo ICP-materials station from 2002–2014 are compared in Table 1.

## 3. Results

The results are presented below, of the cleaning cost and possible savings for sheltered white painted steel facades and modern glass, for the measured air pollution situation and when all the impacting air pollutants were, hypothetically, reduced by 50%, according to Equations (1) to (3). The results for modern glass are presented as the averages calculated through Equations (2) and (3). The separate results for Equations (2) and (3) are shown in Appendix B, Figures A1–A3. The costs were calculated from Equations (4) and (5) for the averages of all the ICP-materials stations, and for the sub-selections of stations noted as urban (U), industrial (I) and rural (R) (Figures 4–6) and Oslo (Figure 7), where the annual average measured values for the needed environmental data were available in the ICP-materials database (Appendix A).

*3.1. Cleaning Costs, and Possible Savings from Reduction in Air Pollution*

Figure 4 shows the average cleaning costs for both materials, calculated to be caused by air pollution higher than the background level, for every year. The background costs, calculated from the background cleaning intervals in Table 2 and Equation (4) (last term), would be 2.9%/year for the cleaning of the white painted steel and 24%/year for the cleaning of the modern glass. These values must be added to the estimated cost over background reported in this work to obtain the values for the estimated total cost.

Figure 4 and Table 3 show a decrease in the cleaning costs due to the air pollution over background level, for the white painted steel and modern glass, for the ICP-materials measurement stations and years.

**Table 3.** The cleaning costs due to the air pollution higher than the background level, as compared to the total cleaning cost, for the mean of the ICP-materials sites after 2002–2005 (Figure 4).

| Material | Cleaning Cost (2002–2005) (%/year [1]) | Cleaning Cost (2011–2014) (%/year [1]) | ΔCost, Start to End (%) |
|---|---|---|---|
| White painted steel (Equation (1)) | 7.0 | 4.0 | −46 |
| Modern glass, mean of (Equations (2) and (3)) | 124 (145, 103) | 95 (117, 72) | −24 (−19, −30) |

[1] % of one cleaning investment, per year during the cleaning interval.

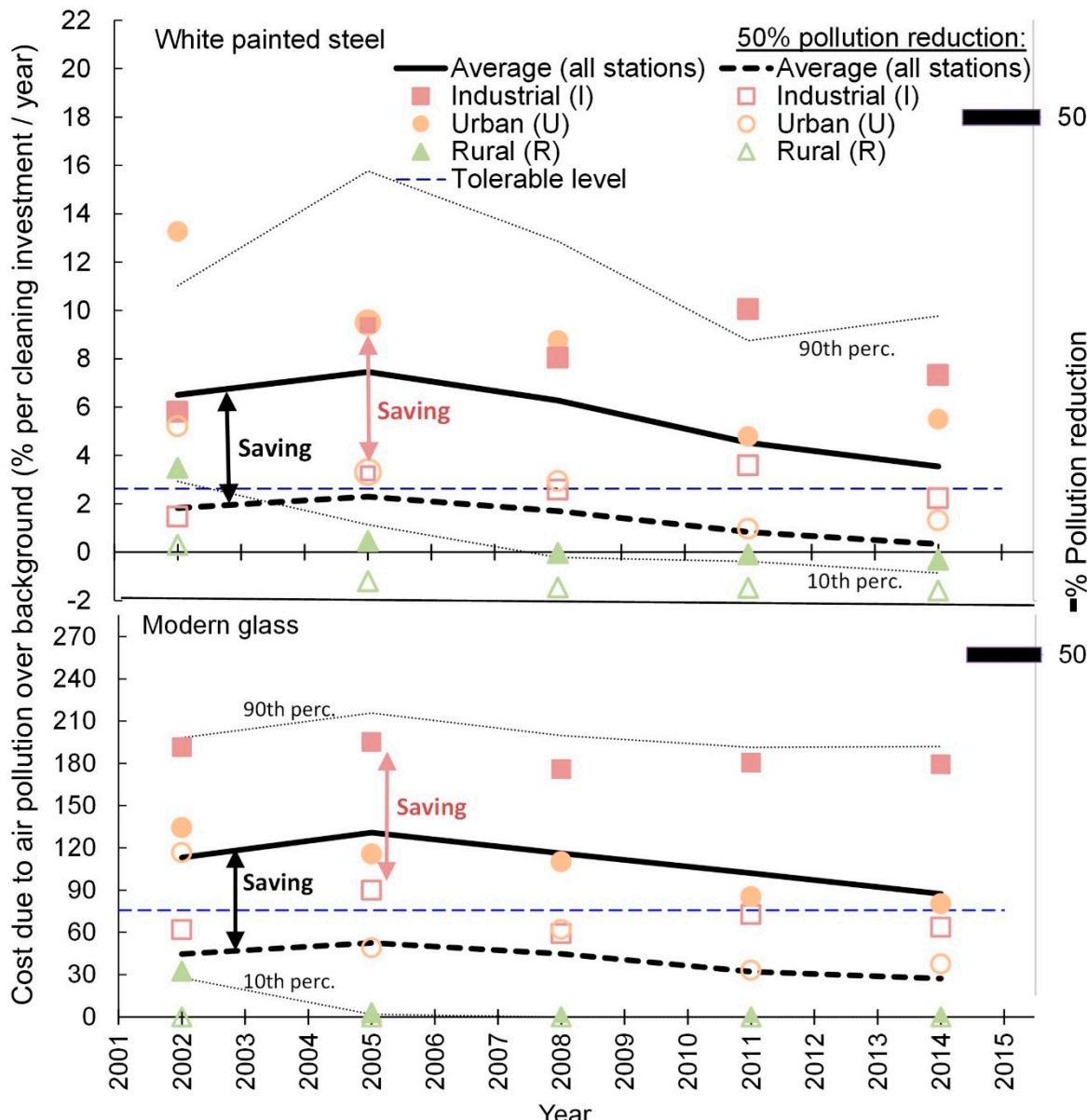

**Figure 4.** The cleaning costs and potential savings due to reduction in air pollution. The calculated average, and 10[th] and 90[th] percentile, of the cleaning costs (% of one cleaning investment, per year during the cleaning interval) caused by air pollution over the background level, at the measured values for the air pollution and that could have been obtained by 50% reduction of the impacting pollutants. The calculations were made with Equations (1) to (5) for all the ICP-materials stations with environmental data, for every year (Appendix A). The values for the urban (U), industrial (I) and rural (R) sub-selections of stations are shown. The savings that could have been obtained by the pollution reduction are given by the differences, as illustrated by the arrows. The savings are independent of the background (Equation (5)). Negative cleaning cost for white painted steel at the rural stations imply lower pollution values and longer cleaning intervals than in the average background (Table 2).

For the white painted steel, the costs (over background) decreased with 46% from 2002–2005 to 2011–2014. For the modern glass, the costs decreased with 24% from 2002–2005 to 2011–2014. The higher values for the costs calculated in 2002–2005 are due to the higher $PM_{10}$ values measured in those years (Table 1 and [28]). The percentage cleaning cost per year of one cleaning investment, due to the air pollution, was found to be ~20 times higher for the modern glass than for the white painted steel, signifying much more frequent need for cleaning of windows, as can also be seen for the

background values in Table 2. A higher cleaning cost than 100%/year for any single station signifies a shorter cleaning interval than one year due to the air pollution over background at that station. The small amount of additional background pollution would increase the need for cleaning and the expected cleaning cost somewhat over this level, and reduce the cleaning interval somewhat. As will be explained below, this direct inverse relationship between costs and cleaning intervals is not similarly valid for the averages for all the stations.

Figure 4 and Table 4 further show some decrease in the average cleaning cost savings for the white painted steel and the modern glass that could have been obtained by 50% reduction in the air pollution, according to Equations (1) to (5), for all the measurement years and ICP-materials stations (Appendix A), as the mean concentration of the air pollutants decreased from 2002–2005 to 2011–2014 (Table 1 and [28]). The relevant environmental parameters were only measured at four sites, one industrial, one urban and two rural sites, in 2002 (see Appendix A). The cost and cost savings (changes) for the site categories were therefore reported from 2005 (rather than 2002) in Table 4.

**Table 4.** The cleaning cost savings (changes) for the sheltered white painted steel and modern glass, due to 50% air pollution reduction at ICP-materials locations. Averages for all stations and years.

| Site Category | Cleaning Cost Saving R, U, I: (2005), All Stations: (2002–2005) (%/year[1]) | Cleaning Cost Saving (2011–2014) (%/year[1]) | ΔCost Saving, Start to End (%) |
|---|---|---|---|
| **White painted steel, Equation (1)** | | | |
| Rural | 2.7 | 2.1 | −20 |
| Urban | 12.2 | 7.9 | −35 |
| Industrial | 11.7 | 11.0 | −6 |
| All stations | 4.9 | 3.5 | −30 |
| **Modern glass: Average of Equations (2) and (3)** | | | |
| Rural | 1.5 (0, 3) | 0 (0, 0) | −100 (0, −100) |
| Urban | 122 (170, 74) | 92 (119, 65) | −24 (-30, −12) |
| Industrial | 105 (129, 82) | 112 (146, 78) | 6 (13, −5) |
| All stations | 75 (94, 56) | 65 (84, 46) | −14 (−11, −19) |

[1] % of one cleaning investment, per year during the cleaning interval.

From 2002−2005 to 2011−2014, the potential savings per cleaning investment (due to 50% pollution reduction) decreased with 30% for the white painted steel, from 4.9%/year to 3.5%/year. For the modern glass, the potential saving per cleaning investment decreased with 14%, from 75% in 2002−2005 to 65% in 2011−2014 (Table 4). The reason for this was the reduction in pollution concentrations (Table 1).

Trends of reduction of cleaning costs and potential savings were calculated for all the ICP-materials station categories (except for the modern glass at the industrial stations, Table 4), although with significant variation between years and between the categories of stations (Table 3, Table 4 and Figure 4). Larger cost savings were, in all instances, found for the industrial and urban stations, compared to the rural stations. The ranking between the industrial and urban stations varied; however, there were larger cost savings at the industrial stations in more years. Generally, there was a decrease in the cost savings over the years for the site categories. However, for the industrial stations, an increase was measured for the modern glass. For the white painted steel, the decrease was significantly smaller than for the other site categories. A very low cost saving was found for modern glass at the rural stations in the start year (2005) sinking to zero cost saving in the end years (2011−2014) (Table 4), reflecting low pollution values.

### 3.2. Comparison with Target Levels and Guidelines

Figure 4 compares the calculated mean values for the costs with the suggested tolerable level (blue lines), for the stations for every year. The considered tolerable cleaning cost, and interval, for the soiling and haze is always constant. The cost saving that could have been obtained by meeting the tolerability target was estimated to have decreased somewhat since 2005, along with the decrease in the air pollution. Figure 4 indicates that a reduction in the air pollution of 50% would, for the mean of all the years and stations, have been sufficient to meet the tolerability target from year 2002 to 2014. This is the case for both the white painted steel and the modern glass. However, this target was not reached in the observed air pollution situation (at the measured values) in any year of measurement. After year 2005 the target would have been reached with gradually less pollution reduction (than 50%).

For both the white painted steel and the modern glass, the tolerability target was reached for the rural average in all measurement years (2002−2014), except for the white painted steel in the first year (2002). For the industrial and urban averages, the target was not reached in any year. For the industrial average, 50% pollution reduction would have been sufficient to reach the target in some, but not all of the years (2005−2014). For the urban averages, the tolerability target would not have been reached by the 50% reduction in pollution in the first years (2002−2008 for the white painted steel and 2002 for the modern glass), but would have been sufficient to reach the target thereafter.

In the dynamic version of the model and diagram (Figure 4), reduction of air pollution from any pollution situation or scenario, for any selection of stations, or single station, can be compared with any given tolerability target to determine if the target was reached or how much pollution reduction had been needed to reach the target.

### 3.3. Indicative Cleaning Intervals, and Possible Increases due to Reduction in Air Pollution

Figures 5 and 6 show the indicative mean cleaning intervals for the white painted steel and the modern glass for the ICP-materials station sites, for the air pollution situation which was measured, and with hypothetical 50% (Figure 5) and continuously increasing (Figure 6) reduction in all the air pollutants, according to Equations (1) to (3). Figure 5 shows results for each year of measurement after 2002, whereas Figure 6 shows mean values from 2002 to 2014. Thus, the means for all the years, of the "% cleaning interval increase that could be obtained by 50% pollution reduction", shown in Figure 5, are also given at 50% reduction in the pollution in Figure 6. The calculations applied the proposed tolerable soiling of the white painted steel, giving 35% relative loss of reflectance, and a value of 3% for the tolerable haze of modern glass (see Section 2.4). It should be noted that a near 100% reduction in air pollution is only hypothetical, as this would be below the natural background level evaluated in this work as the 10th percentile of measured values (See Section 2.4).

The results in Figure 5 indicate increase in cleaning intervals from 2002 to 2014 (based on the linear regression lines given in the figure), as a mean for the ICP-materials sites. For the white painted steel, the increase is from 12 to 24 years (i.e. 100%), and for modern glass it is from 1.2 to 2.0 years (i.e., 70%). The additional increases obtained by hypothetical 50% reduction in air pollution, was indicated to be 100% for the white painted steel. For the modern glass, it was indicated to be 115% in 2002, sinking to 45% in average for the years 2005 to 2014.

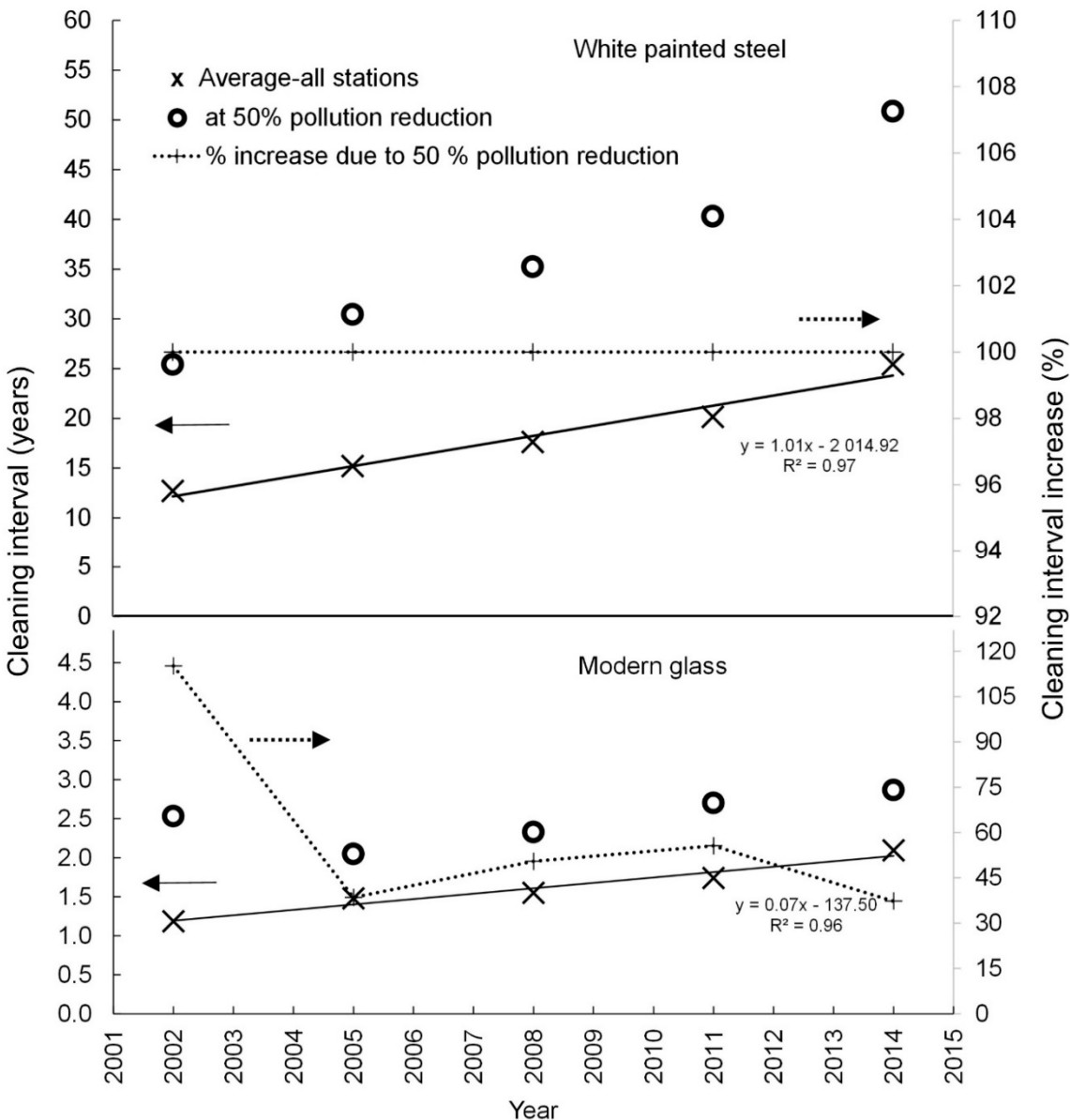

**Figure 5.** The cleaning intervals. Indicated mean intervals before recommended cleaning (years), and increase in the cleaning intervals (%), of the white painted steel and the modern glass due to hypothetical 50% reduction in all the air pollutants, according to Equations (1) to (3), for the ICP-materials sites after 2002. Tolerable loss of reflectance before cleaning of 35% (for white painted steel) and a tolerable annual haze of 3% (for the modern glass) were used. The "% cleaning interval increase" is the given difference between the cleaning interval at 50% pollution reduction and the average values for all the stations.

For single stations and years, a cleaning cost above the tolerable level signifies that the cleaning interval is shorter than tolerated (recommended) according to Table 2. It is worth noting that there is no such direct correspondence between the values for the averages in Figures 4 and 5. The average costs over background given in Figure 4 were over the tolerable level, while the mean cleaning intervals seen in Figure 5 were still usually longer than the tolerable cleaning intervals, of one year for modern glass and 18 years for white painted steel, given in Table 2. The reason for this is the larger weight given in the mean to the high values of cleaning costs and times. Thus, average cost can exceed a threshold due to high costs in polluted locations, while the average cleaning interval is still longer than the tolerability limit, due to long intervals in clean locations.

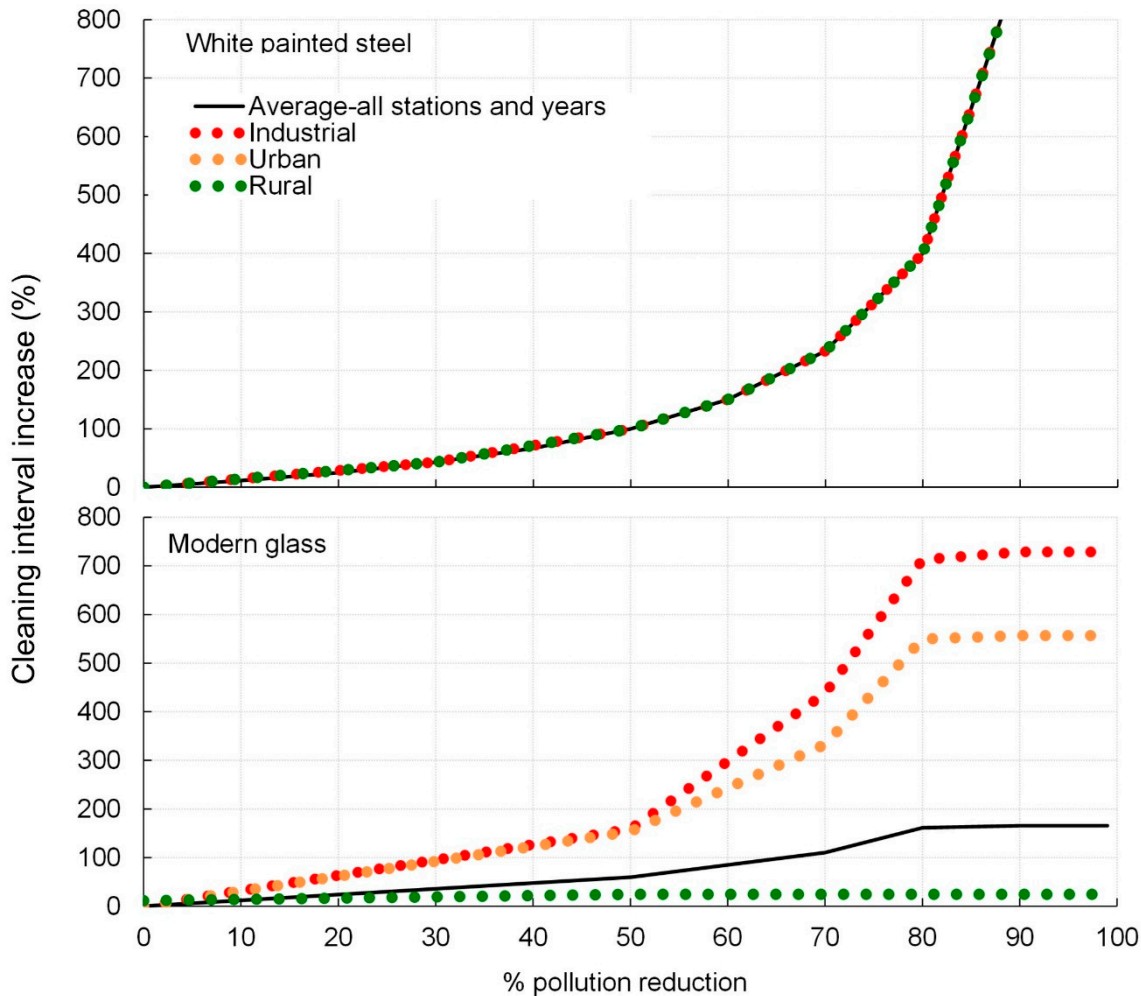

**Figure 6.** The cleaning interval increases due to a hypothetical reduction in air pollution. The mean per cent increase in the cleaning interval for the white painted steel and the modern glass from 2002 to 2014, calculated for the ICP-materials sites and for the sub-selections of industrial, urban and rural sites, which could be obtained by increasing percentages reduction of all the effective air pollutants. The calculations were made for a tolerable soiling of the white painted steel giving 35% relative loss of reflectance, and a tolerable haze for modern glass of 3%.

Figure 6 shows that for the white painted steel, the cleaning interval expectancy increases to the double duration at 50% pollution reduction, then to five times the duration at 80%. With even more reduction in the air pollution, a sharp increase in the cleaning interval is expected. Due to the model formulation (Equation (1)) the expected relative (%) increase in the cleaning interval expectancy does not depend on the concentration level of $PM_{10}$ and is therefore the same for the rural, urban and industrial stations. The average increase in the expected cleaning interval for the modern glass that could be obtained by the reduction in the air pollution was calculated to be somewhat less than for the white painted steel until 50% pollution reduction, when it was 59%, then much less at 80% pollution reduction, when it was 162%. However, the % increase in the expected cleaning interval for the modern glass was higher at the urban and industrial stations, and lower at the rural stations, than for the white painted steel. At high pollution reduction (> 80%), the cleaning interval expectancy for the modern glass approached the constant determined by the maximum glass cleaning interval of ~4 years (1500 days) used in the modelling.

### 3.4. Example of Mapping of Possible Savings from Reduction in Air Pollution for Oslo, Norway

Figure 7 shows the example of the mapping of the savings in the cleaning costs for the white painted steel surfaces and the modern glass that could probably have been obtained by 50% reduction in air pollution, for the years 2003–2014. Resulting in less soiling of the painted surfaces and glass in Oslo, Norway.

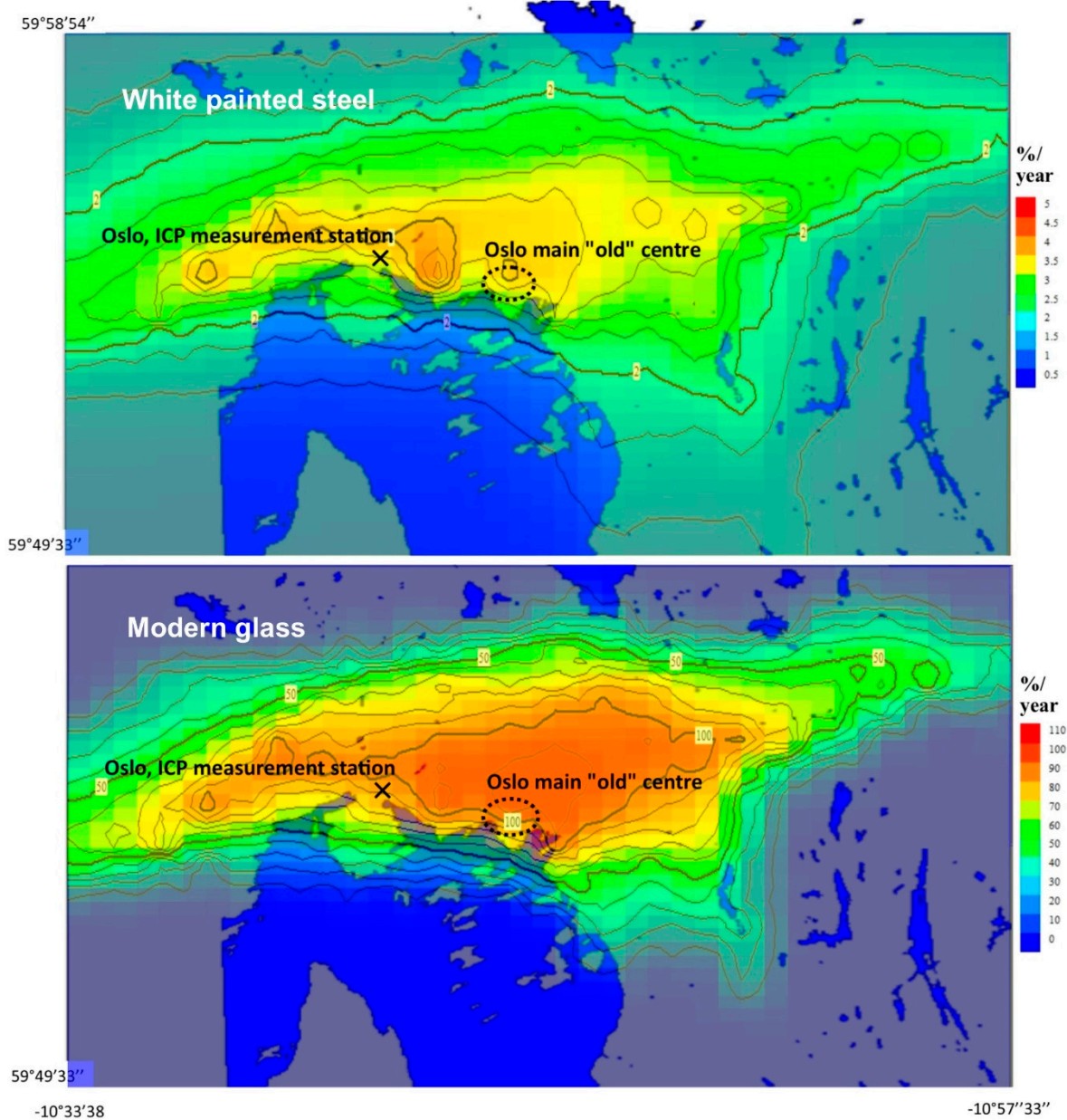

**Figure 7.** The mapping of possible savings from reduction in air pollution for Oslo, Norway. Expected % cleaning cost savings per year for the white painted steel due to soiling, and the modern window glass due to haze, in positions sheltered from precipitation, in Oslo, Norway, that could probably have been obtained by hypothetically reducing the air pollution by 50% after year 2002. The location of the ICP-materials station in Oslo is shown. The highest potential cost savings were in the city centre. %/year = % of one cleaning investment, per year during the cleaning interval.

Validation was carried out by comparing the results from the mapping for the modern glass with the calculated values for the % saving in cost by Equations (2) and (3), for the ICP-materials station in Oslo (location shown on map), for the years 2002 to 2014. $PM_{10}$ was not measured at the

Oslo ICP-materials station. Therefore, the same value as was used in the mapping (21.3 µg/m$^3$) was used in the validation (Table 1). The environmental values, which were measured at the Oslo ICP-station in the years 2002–2014 and used as input to the estimations of the cost savings for the Oslo ICP-materials site, and the values applied in the mapping are compared in Table 1. The soiling of white painted steel calculated by Equation (1) only depends on a variation in PM$_{10}$. For the white painted steel, the mapping therefore directly represents an approximated value of PM$_{10}$ = 21.3 µg/m$^3$ at the ICP-materials urban background station in Oslo and the input model variation. For modern glass (Equations (2)–(3)), the calculated values for the cost saving depends in addition on the concentration values for SO$_2$ and NO$_2$ (Table 1). The value for the possible saving in the cleaning cost for modern glass for the location of the Oslo ICP-materials station, obtained from the mapping, was 94%/year, as compared to a value of 90%/year calculated from the ICP-materials station measurement values. This gives reasonable confidence to the mapping results.

The mapping shows the amounts and variation in the probable savings in cleaning cost for surfaces of the white painted steel and the modern window glass, sheltered from precipitation, obtainable by air pollution reduction of in the city. This may change in the future with an expected further reduction in the air pollution concentrations, due to especially electrification of transport and expected less pollution from wood burning stoves.

## 4. Discussion

Some aspects related to the validity of the results for the cleaning costs will be discussed. The reported results for the cleaning costs and savings that could have been obtained for the white painted steel and the modern glass, sheltered from precipitation, will then be compared with previous reported results for atmospheric weathering costs. This will allow a broader assessment of the maintenance cost and savings, including a discussion about absolute prices and costs.

### 4.1. Uncertainties in Costs Calculations

This work presents calculations of cleaning costs and potential savings due to hypothetical air pollution reduction in recent years and cleaning intervals for white painted steel and modern glass, by using trends for European stations in locations with different environments. The results are not reported as "European averages". That would require a discussion about representativeness, which is not given here. Urban (12) and rural (eight) sites are overrepresented as compared to industrial (three) sites. The ICP-materials project distinguishes between these location-categories to allow assessment of variation in exposure, soiling and corrosion depending on main general differences between the locations. The time series estimates (since 2002) provides information about recent trends in cleaning costs. Long-range transport of air pollution and dispersion over large areas generally affect rural locations more compared to emission from single pollution sources. Measurement values at urban stations are typically more affected by emissions close to the stations, for example from traffic. Such emissions may be unrelated to general pollution trends in cities. Most of the urban ICP-materials stations were located in the "urban background". The exposure at the stations would generally be not mainly from one source, such as a local road, but from a larger area of the city. How the few (3) industrial stations represent "industrial" stations in general is more uncertain. The soiling and calculated cleaning costs were high at the industrial sites, but would be more dependent on the selection of each site. Since 2002, there was a significant increase in the number of stations in the ICP-materials network, which measured the environmental parameters influencing the soiling (Section 2.3, "Data", and Appendix A). The basis for the evaluation of trends, including 2002 and 2005, is weak.

The savings in cleaning costs due to reduction in air pollution were calculated (by Equations (1) to (5)) for hypothetical periods into the future with the same air pollution levels as in the measurement years. The calculated cleaning intervals for white painted steel were often several decennia; with decreasing values of PM$_{10}$ since 2002 in Europe [46], as seen with the ICP-materials stations (Table 1),

and possible future decreases, these calculations could exaggerate the future integrated cleaning costs from any one year. The future possible savings in cleaning costs would probably be somewhat lower for the white painted steel than reported in Figure 4 and Table A1. For the modern glass, for which the calculated cleaning interval is always less than four years, this effect would be insignificant (Figure 5).

The development of the soiling function for the white painted steel (1) was based on one year of experimental data. It is uncertain how well the first year development of the soiling describes the further progress of the soiling over many years [10]. This uncertainty would be larger for the cleaner atmospheres with long cleaning intervals. It is difficult to evaluate the importance of the soiling and need of cleaning at low particulate pollution levels. The uncertainty in the dose-response functions (Equations (1) to (3)) may be higher at low $PM_{10}$ values. This is a small part of their statistical basis.

A white smooth surface, like white painted steel, could be considered an indicator for soiling and the respective cleaning costs. However, such indication of soiling and costs should not be interpreted to directly represent other different surfaces. Soiling is expected to be most noticeable on a white surface. The soiling will also depend on the retention of particles on the surface, which will depend on characteristics such as its roughness. The calculated values for the reported cleaning costs for white painted steel may be in the upper range, for sensitive cases. Longer cleaning intervals could be acceptable for different surfaces where the soiling is less immediately apparent. The cleaning costs would then be less.

Tolerable levels of 35% relative loss of reflectance on white painted steel and 3% haze on modern glass were used as the criterion for the time of the cleaning, and the cleaning intervals. It is uncertain how well this describes the cleaning practice in different cases. Soiling is a complex phenomenon depending on the properties of the air pollutants and surfaces, and on the atmospheric conditions. The physical and perceived soiling of, for example, rough and dark surfaces and facades and windows exposed to rain-washing, is expected to be significantly different from the indicator surfaces evaluated in this work. The soiling of dark surfaces may be considered less of a nuisance. The rain-washing of facades can create patterns, which cannot be evaluated according to some overall soiling effect such as the general loss of reflectance. The rain-washing of windows could reduce the perceived need for cleaning, and thus increase cleaning intervals. Cleaning intervals are likely to be affected by the amount, colour and pattern of soiling and by public attitudes [47,48]. Cleaning intervals for windows that attract special attention could be much more frequent than one year, giving higher cleaning cost. These could be for example in cultural heritage or commercial prestige buildings or shopping areas. For example, the glass Pyramid at the Louvre Museum in Paris is cleaned once per month using a robot. It has been found that in this area (i.e., the centre of Paris), the haze on a float glass in unsheltered conditions was 1% after 1 month of exposure [49]. Shop windows or business buildings are usually cleaned more frequently (every week or every 1 or 2 months) than other windows and facades. The realized cost would also depend on how the cleaning practices are related to the soiling. The performed calculations assumed that the cleaning of surfaces is a direct response to the level of soiling. It seems safe to state that cleaning is in general a response to soiling, but it is probably not the only reason for variation in the cleaning frequency. The cleaning may for example be carried out at certain intervals according to some maintenance plan, with more or less disregard to the amount of soiling. Economic factors would clearly also be important.

The presented results should be considered as indications, probably in the upper range of the empirical variation, of the cleaning costs and the possible costs savings from pollution reductions. For windows exposed to rain and non-smooth and non-white surfaces, the cleaning intervals are expected to be longer, and the cost and cost savings lower. What is tolerable soiling and haze will certainly be evaluated differently in different concrete situations, and the circumstances for the cleaning will vary. This will give variations in the cleaning frequency at and between locations, including the ICP-materials locations, different from the results presented in this work.

### 4.2. Comparison with Corrosion Costs

In the present multi-pollutant situation, the cost related to the cleaning of facades and structures may be higher than maintenance costs due to atmospheric weathering and corrosion [15,17,50]. Thus, the soiling should be given as much attention as the weathering. Comparison can be made with results obtained in previous work for the maintenance cost due to atmospheric weathering.

The costs for maintenance of zinc and Portland limestone due to air pollution over the background level, relative to the total maintenance cost, for the average of the ICP-materials stations, have been reported [11]. For zinc, the cost was found to be 0.24%/year (the unit %/year equals here as through the paper: % of one cleaning investment, per year during the cleaning interval), from 2008–2014. For Portland limestone, the cost was found to be 39% of one maintenance investment, from 2011–2014. The cost per year depends on the amount of weathering that is tolerated before maintenance is implemented. This tolerable amount of weathering is more difficult to assess for the less uniformly weathered surfaces of limestone than zinc [11,12]. A recommended value of 100 μm "tolerable corrosion of limestone/marble aged ornament before action" [18], was used to calculate a lifetime between maintenance for Portland limestone ornament of 44 years, for the average of the ICP-materials locations in 2014 [11]. This, then, corresponds to a maintenance cost due to air pollution over the background level of ~1%/year.

It was found that 50% reduction in air pollution at ICP-materials locations would probably reduce maintenance costs due to atmospheric weathering of zinc monument surfaces with about 0.1% per year, and of Portland limestone ornament surfaces with about 10% of one maintenance investment, as an average for the stations [11]. For Portland limestone, this corresponds to a value of ~0.2%/year, for the lifetime of 44 years between maintenance. The tolerable recession before maintenance of less decorated surfaces would probably be higher than for limestone ornament. Rable et al. [24] suggested a critical thickness loss of natural stone before maintenance or repair of 4 mm (this was used in the ExternE project [51]). This is a factor of 40 larger than suggested for limestone ornament, and illustrates a large difference in evaluated wear tolerances for ornaments and general surfaces. For the ICP-materials locations in 2014 and the applied DRFs, this would imply a lifetime between maintenance of 1760 years, and a cost due to air pollution over the background level of 0.025%/year, and saving of 0.005%/year due to 50% reduction in air pollution. This seems to merely illustrate low present pollution costs in situations with large tolerances. The deterioration over such long time periods would depend on factors, which are beyond the evaluation in this work, as is evident from archaeological building remains. The industrial air pollution since about year 1800 has however had a disproportionate weathering impact in many European locations [52]. In the comparisons made below the reported values for the general maintenance costs and savings of "limestone ornament", should be considered to be in the upper range of air pollution weathering costs for this material.

In this work, it was estimated that 50% reduction in air pollution would have resulted in savings in cleaning cost for white painted steel surfaces sinking from about 5%/year in 2002–2005 to 3.5 %/year in 2011–2014, and for modern glass sinking from about 75%/year in 2002–2005 to 65%/year in 2011–2014 (Figure 5, Table 4).

A summary of the results for relative cleaning costs and costs savings, reported in this work, and for the surface maintenance of limestone and zinc [11], are given in Table 5. The table further shows results for the calculation of absolute costs and cost savings, based on the relative values, as depending on the Euro cost for one cleaning or maintenance intervention. When sorting the European average conservation and renovation costs related to cultural heritage (for the UK, Czech republic and Norway in 2006, Euro/m$^2$) reported for six relevant maintenance categories (roof envelope cost, non-plastered masonry, glass walls, plastered facades, window cleaning as part of supporting works and sculptures and sculptural items; costs items judged to mainly be different from general surface maintenance of limestone due to soiling, such as wood work, replica production, complex restoration and general scaffolding were removed from the calculation) [53], into three new categories: 1. window cleaning (three entries); 2. facade and surface cleaning (12 entries) and; 3. facade and surface maintenance

(84 entries). By averaging the cost within each new category, one obtains for the new categories (1, 2, 3–4, Table 5) a maintenance cost of 2, 60 and 85 Euro/m$^2$, respectively. The third category, "facade and surface maintenance" was used for both zinc and Portland limestone as the variation within the category depending more on the type of work than type of surface material. There were relatively few entries for metals (7), and a separate category for "metals" (zinc) would, due to variation in cost between these entries, have produced an unrealistic difference between the stone- and metalwork. The empirical cost values (Euro/m$^2$) were then adjusted with the change in the European Union Labour Cost Index from 2006 to 2018, of 30 % [54], to obtain the values reported in Table 5. By multiplying these empirical cost values (Euro/m$^2$) with the mean relative costs and cost savings in Table 5 (Equations (4) and (5)), the approximations of the averaged present (2018) absolute costs due to air pollution and costs savings from reduction in air pollution, given in Table 5, were obtained. As was noted in Section 3.3, it should be stressed again that the average absolute yearly costs over background for all the stations, reported in Table 5, are not simple divisions of the total empirical costs by the maintenance interval. This would be the case for the absolute yearly cost for any single station when including the background cost, but not so for the averages of all the stations. In Table 5, this is coincidentally the case for the white painted steel. If, however, the background cost is included with the average stations absolute cost for the cleaning of the white painted steel, it would be 5.4 Euro/m$^2$·year.

**Table 5.** Approximation of "present" (2018) cleaning (row 1 and 2) and maintenance (row 3 and 4) costs over background air pollution, and savings due to 50 % reduction in the air pollution, calculated for averages of the ICP-materials stations. The maintenance interval assessment was based on measurement values from recent years before 2014.

| Maintenance Category | Empirical Cost (E) [2] (Euro/m$^2$) | Maintenance Interval [3] (~years) | Relative Cost [4] (%/Year [7]) | Absolute Cost [6] (Euro/m$^2$ Year) | Relative Cost Saving [5] (%/Year [7]) | Absolute Cost Saving [6] (Euro/m$^2$ Year) |
|---|---|---|---|---|---|---|
| 1. Cleaning of modern glass | 2.6 | 2 | 95 | 2.5 | 65 | 1.7 |
| 2. Cleaning of white painted steel | 78 | 25 | 4.0 | 3.1 | 3.5 | 2.7 |
| 3. Surface maintenance (limestone [1]) | 110 | 50 | 1 | 1.1 | 0.2 | 0.2 |
| 4. Surface maintenance (zinc [1]) | 110 | 200 | 0.24 | 0.3 | 0.1 | 0.1 |

[1] See [11]. [2] The empirical costs are for the general categories of: Row 1. window cleaning, Row 2. façade and surface cleaning and; Rows 3 and 4. facade and surface maintenance. See discussion above Table 5. [3] For the cleaning, Rows 1 and 2: See Figure 5. For the surface maintenance, Rows 3 and 4: see [11]. [4] For the cleaning, Rows 1 and 2: See Table 3: "Mean of the ICP-materials sites (2011–2014)". For the surface maintenance, Rows 3 and 4: see Discussion. [5] For the cleaning, Rows 1 and 2: See "Averages of all ICP-materials stations (2011–2014)", Table 4. For the surface maintenance, Rows 3 and 4: see Discussion. [6] According to calculation from values given in Table 5. See Discussion. [7] %/year = % of one cleaning investment, per year during the cleaning interval.

Table 5 shows differences in possible relative cost, and cost savings from reduction in air pollution (with 50%). They are more than one order of magnitude higher for the cleaning of sheltered glass (cost of 95%/year and cost savings of 65%/year) than for the cleaning of white painted steel surfaces (cost of 4.0 and cost savings of 3.5%/year). Then they are about five times more than one order of magnitude higher than the maintenance cost for the limestone ornament and zinc surfaces due to atmospheric chemical weathering (cost of 4.0 and cost savings of 3.5%/year, as compared to cost of 1–0.24 and cost savings of 0.2–0.1%/year).

The cleaning costs for the smooth white painted surfaces are probably lower than those for the mostly non-painted surfaces related to cultural heritage [53]. The entries in Reference [53] include some high cost operations like "cleaning of severely damaged stone including desalination" and "cleaning of decorated lime plaster (fresco)". An empirical cost of ~50 Euro/m$^2$ (rather than the 78 Euro/m$^2$ in Table 5) would give near similar results for the absolute costs for the glass and the white painted steel surfaces. Cleaning costs for building façades in Sweden and France ranging from of 5 to 18 Euro/m$^2$, and an average value for cleaning of painted surfaces in Sweden, of ~6 Euro/m$^2$ have been reported [24,55]. As more expensive maintenance than cleaning, such as repainting, may in some

instances be the response to soiling, they evaluated the present cost due to façade soiling in Sweden to be about 30 (15–45) Euro/m$^2$, approaching the value in Reference [53]. However, significantly shorter cleaning intervals were reported in Reference [55] (3–10 years as compared to the 25 years calculated for the average of the ICP-materials sites from Equation (1), Table 5). This illustrates uncertainties in such calculations, which would reflect also variation in the surface materials, experienced costs and practices.

Thus, the lower values in the cleaning cost range are more likely to represent average costs for both the sheltered white painted surfaces and glass. This calculation then indicates that for the sheltered window glass and white painted surfaces, the average present (2018) cleaning cost due to air pollution over background may be about 2.5 Euro/m$^2$·year, and the cleaning cost savings that could be obtained by reducing air pollution in Europe with 50%, may be about 1.5 Euro/m$^2$·year. It is further indicated that the absolute cleaning costs over background (Euro/m$^2$·year) are possibly somewhat higher (~two times) than the maintenance cost due atmospheric weathering and corrosion of limestone facades/ornament, but maybe nearly one order of magnitude higher than for zinc monument surfaces. It should be considered here that the variation in maintenance cost for limestone surfaces is expected to be large depending on their characteristics and that is could be considerably higher for ornamented surfaces than the average value (of 110 Euro/m$^2$) reported in Table 5. It is also indicated that the savings that could be obtained by 50% air pollution reduction are significantly, possibly one order of magnitude, higher for the cleaning than for the surface maintenance. These cleaning costs do not include the amenity loss or take discounting into consideration. The amenity loss has been found to be approximately equal to the renovation cost [24]. Discounting would mean the distribution of a maintenance-cleaning investment on decreasing annual costs with time according to a discount rate, rather than simply dividing it by the maintenance-cleaning interval to obtain the annual cost.

Large existing differences in physical properties of facades and surfaces and in attitudes and cleaning practices are expected to significantly influence cleaning intervals and costs. In the approach of this work, different factors other than the physical soiling were treated as constants, which would not affect the calculated differences in the cleaning costs between years or pollution scenarios. The presentation of cleaning costs and savings calculated by this standardized method assures comparability between locations and makes averaging possible. To assess the cleaning intervals and cost for single cases and locations, as compared to those reported here, one should evaluate differences in the properties and exposure situations, from those of the white painted steel and the modern glass sheltered from precipitation, and differences in the perception of tolerable soiling and in cleaning practices. The reported assessment of the cleaning costs of facades and window glass as compared with weathering and corrosion costs, can further allow comparison with total maintenance cost for buildings, and thus, assessment of the relative importance of the air pollution for total maintenance costs.

## 5. Conclusions

It was found that the present (2018) average cleaning costs due to air pollution over background for sheltered windows and white painted steel surfaces in Europe was probably about 2.5 Euro/m$^2$·year. It was further found that a hypothetical reduction in the air pollution in Europe with 50%, would probably give savings in these cleaning costs of about 1.5 Euro/m$^2$·year. These annual cleaning cost due to air pollution are somewhat higher (~2 times) than the previously reported maintenance costs for Portland limestone ornament surfaces, but about one order of magnitude higher than the costs reported for zinc monument surfaces. The cost savings (Euro/m$^2$·year) that could have been obtained from 50% reduction in air pollution were found to be about one order of magnitude higher for the cleaning than for maintenance of the ornament (Portland limestone) and zinc monument surfaces.

The reported result indicates a respective possible increase, due to the hypothetical 50% air pollution reduction, in the future cleaning interval for sheltered white painted smooth facades and surfaces, and for modern glass, of between 50 and 100%, with a larger increase for the white painted

surfaces than for the modern glass. The future cost savings relative to one present cleaning action, due to this reduction measure, would probably be about 3 %/year (calculated to sink from about 5%/year in 2002–2005 to 3.5%/year in 2011–2014) for the sheltered white painted steel facades, and about 50%/year (calculated to sink from about 75%/year in 2002–2005 to 65%/year in 2011–2014) for the modern window glass.

The average increase in the theoretical cleaning interval over the period from 2002–2005 until 2011–2014 at the ICP-materials locations, due to the measured reduction in air pollution, was found to be: for the white painted steel in a position sheltered from precipitation, about 100% (increasing from 12 to 24 years), representing reductions in cleaning costs from 7%/year to 4%/year; for the modern glass, about 65% (increasing from 0.85 to 1.3 years), representing reductions in cleaning cost from 124%/year to 95%/year (the unit %/year equals here as through the paper: % of one cleaning investment, per year during the cleaning interval).

However, large differences in the surface properties, perceived need for cleaning and realized cleaning costs are expected. Many facades and surfaces are less bright than white painted steel and many windows are exposed to rain, giving longer expected cleaning intervals and lower cost. The cleaning intervals for, for example, highly valued surfaces of cultural heritage or shopping display windows would often be much shorter, giving shorter intervals and higher cost. The total calculated costs savings from reduction of air pollution would have been and could potentially be very significant, considering the total areas of windows and white painted (and other) facades. The significance of soiling and cleaning cost should in every specific case be evaluated in the context of the total and combined environmental loads on a built structure and the related maintenance opportunities and costs. In some cases, soiling can be an important impact and cost; in other instances, it will be of little importance.

**Author Contributions:** All three authors, T.G., A.V.-C. and J.T., have been involved in the ICP-materials project (http://www.corr-institute.se/icp-materials), within which measurements of soiling of glass and of environmental parameters took place, and where the basis for the cost assessments methodology was developed. A.V.-C. provided dose-response functions for glass and their interpretation. T.G. supplied the main estimations and developed the paper. J.T. especially assessed targets and guidelines to be included in the assessment and gave input to the discussion about empirical (Euro) cost for maintenance. All the authors did a final review. Conceptualization: T.G.; methodology, J.T., T.G. and A.V.-C.; software, T.G.; validation, T.G.; formal analysis, T.G.; investigation, A.V.-C., T.G. and J.T.; resources, T.G. and A.V.-C.; data curation, J.T., T.G. and A.V.-C.; writing—original draft preparation, T.G.; writing—review and editing, T.G., A.V.-C. and J.T.; visualization, T.G.; supervision, T.G.; project administration, J.T. and T.G.; funding acquisition, T.G.

**Funding:** This research was performed with support from the Norwegian Environment Agency, from NILU - the Norwegian Institute for Air Research as partner in ICP-materials, from the French Environment & Energy Management Agency (ADEME) and from the Swedish Environmental Protection Agency.

**Acknowledgments:** The basis for this work is the 30 years of field work, analysis and reporting by the partners in the International Co-operative Programme on Effects on Materials, including Historic and Cultural Monuments (ICP-materials). The ICP-materials programme was started in 1985 to provide a scientific basis for protocols and regulations to be developed within the Convention on Long-range Transboundary Air Pollution under the United Nations Economic Commission for Europe (UNECE). We thank all present and past ICP-materials partners for their invaluable work over all these years, and for the particle and soiling measurements, which started around year 2000, and which today make works like this possible.

**Conflicts of Interest:** The authors declare no conflict of interest.

## Appendix A

This appendix contains tables with the estimated values for the cleaning cost and potential cost savings due to 50% hypothetical reduction in air pollution, for sheltered white painted steel and modern glass, for all the single stations and years of measurements of the environmental parameters needed for the calculations, in the ICP-materials project station network. The results for modern glass are reported individually for Equations (2) and (3). The averages from calculation by Equations (2) and (3) are given in the main text. The results reported in the Appendix tables represent the data availability at the ICP-materials stations [28].

**Table A1.** The cleaning cost for white painted steel exposed outdoors in sheltered position, due to $PM_{10}$ concentrations over background (Equations (1) and (4)), and cleaning cost savings (% of one cleaning investment, per year during the cleaning interval = %/year) due to 50% reduction in $PM_{10}$ values, for the reported stations and years (Equation (5)). Negative values indicate cleaning costs lower than background values. The table represents the data availability for the ICP-materials stations and years. The $PM_{10}$ input values to the calculations are, for the years from 2002 until 2011, reported in Reference [28], and for 2014 reported in Reference [36]. (U) = Urban, (I) = Industrial, (R) = Rural.

| Cleaning Cost: | Due to $PM_{10}$ over Background (%/year) | | | | | Saving due to 50% Reduction in $PM_{10}$ (%/year) | | | | |
|---|---|---|---|---|---|---|---|---|---|---|
| Station Year: | 2002 | 2005 | 2008 | 2011 | 2014 | 2002 | 2005 | 2008 | 2011 | 2014 |
| 1 Prague (U) | | 3.9 | 3.6 | 6.2 | 4.3 | | 3.4 | 3.3 | 4.5 | 3.6 |
| 3 Kopisty (I) | | 4.0 | 6.7 | 7.7 | 5.8 | | 3.4 | 4.8 | 5.3 | 4.4 |
| 7 Waldhof-Langenbrügge (R) | 4.9 | | | | | 3.9 | | | | |
| 10 Bottrop (I) | 5.8 | 7.6 | 6.5 | 5.9 | 5.0 | 4.3 | 5.2 | 4.7 | 4.4 | 4.0 |
| 13 Rome (U) | | | | | 5.1 | | | | | 4.0 |
| 14 Casaccia (R) | | | | | 0.6 | | | | | 1.7 |
| 15 Milan (U) | | | | | 9.6 | | | | | 6.2 |
| 23 Birkenes (R) | | −0.7 | −1.0 | −0.9 | −1.2 | | 1.1 | 1.0 | 1.0 | 0.8 |
| 24 Stockholm south (U) | | | | 1.7 | 1.2 | | | | 2.3 | 2.1 |
| 26 Aspvreten (R) | | | | −0.5 | −0.2 | | | | 1.2 | 1.3 |
| 31 Madrid (U) | | | 2.5 | 2.3 | 2.4 | | | 2.7 | 2.6 | 2.6 |
| 33 Toledo (R) | | | 1.1 | 1.2 | 1.4 | | | 2.0 | 2.0 | 2.1 |
| 35 Lahemaa (R) | | 1.6 | −0.6 | | −0.7 | | 2.2 | 1.2 | | 1.1 |
| 40 Paris (U) | | 4.7 | 6.8 | | | | 3.8 | 4.8 | | |
| 41 Berlin (U) | 13.3 | | 9.7 | 6.5 | 6.8 | 8.1 | | 6.3 | 4.7 | 4.8 |
| 44 Svanvik (R) | | | | | −1.6 | | | | | 0.6 |
| 45 Chaumont (R) | 2.1 | | 0.3 | −0.1 | −0.4 | 2.5 | | 1.6 | 1.4 | 1.2 |
| 50 Katowice (I) | | 16.6 | 10.9 | 16.5 | 11.1 | | 9.7 | 6.9 | 9.7 | 7.0 |
| 51 Athens (U) | | 15.6 | 14.1 | 9.2 | 10.2 | | 9.2 | 8.5 | 6.0 | 6.5 |
| 52 Riga (U) | | 13.9 | 10.6 | | | | 8.4 | 6.7 | | |
| 53 Vienna (U) | | | 4.7 | 4.7 | 4.3 | | | 3.8 | 3.8 | 3.6 |
| 54 Sofia (U) | | 17.9 | | | | | 10.4 | | | |
| 55 Saint-Petersburg (U) | | | | 2.9 | | | | | 2.9 | |

**Table A2.** The cleaning cost for modern glass surfaces exposed outdoors in sheltered position due to air pollution over background (% of one cleaning investment, per year during the cleaning interval = %/year) (Equations (2), (3) and (4)), for the reported stations and years. The averages reported in Figure 4, were calculated as the mean of the averages over the stations for each year in the table, for Equations (2) and (3). The cells with reported values represent the data availability for the ICP-materials stations and measurement years. The values for the measured environmental variables used as input to the calculations are, for the years from 2002 until 2011, reported in Reference [28], and for 2014 reported in Reference [36]. (U) = Urban, (I) = Industrial, (R) = Rural.

| | Cleaning Cost Due to Air Pollution over Background (%/year) | | | | | | | | | |
|---|---|---|---|---|---|---|---|---|---|---|
| | Equation (2) | | | | | Equation (3) | | | | |
| Station Year: | 2002 | 2005 | 2008 | 2011 | 2014 | 2002 | 2005 | 2008 | 2011 | 2014 |
| 1 Prague (U) | | 226 | 174 | 238 | 198 | | 127 | 104 | 129 | 106 |
| 3 Kopisty (I) | | 158 | 180 | 227 | 267 | | 131 | 128 | 152 | 149 |
| 7 Waldhof-Langenbrügge (R) | 63 | | | | | 53 | | | | |
| 10 Bottrop (I) | 237 | 241 | 221 | 181 | 163 | 146 | 151 | 132 | 117 | 109 |
| 13 Rome (U) | | | | | 163 | | | | | 88 |
| 14 Casaccia (R) | | | | | 0 | | | | | 0 |
| 15 Milan (U) | | | | | 233 | | | | | 141 |
| 23 Birkenes (R) | | 0 | 0 | 0 | 0 | | 0 | 0 | 0 | 0 |
| 24 Stockholm south (U) | | | 23 | 24 | | | | 12 | 6 | |
| 26 Aspvreten (R) | | | 0 | 0 | | | | 0 | 0 | |
| 31 Madrid (U) | | | 56 | 128 | 110 | | | 52 | 69 | 62 |
| 33 Toledo (R) | | | 0 | 0 | 0 | | | 0 | 0 | 0 |
| 35 Lahemaa (R) | | 0 | 0 | | 0 | | 6 | 0 | | 0 |
| 40 Paris (U) | | 214 | 215 | | | | 117 | 118 | | |
| 41 Berlin (U) | 219 | | 226 | 231 | 218 | 173 | | 138 | 121 | 116 |
| 44 Svanvik (R) | | | | | 0 | | | | | 0 |
| 45 Chaumont (R) | 17 | | 0 | 0 | 0 | 12 | | 0 | 0 | 0 |

**Table A2.** *Cont.*

| | Cleaning Cost Due to Air Pollution over Background (%/year) | | | | | | | | | |
| | Equation (2) | | | | | Equation (3) | | | | |
| Station Year: | 2002 | 2005 | 2008 | 2011 | 2014 | 2002 | 2005 | 2008 | 2011 | 2014 |
|---|---|---|---|---|---|---|---|---|---|---|
| 50 Katowice (I) | | 256 | 236 | 219 | 233 | | 235 | 157 | 187 | 155 |
| 51 Athens (U) | | | 226 | 232 | 233 | | | 184 | 147 | 161 |
| 52 Riga (U) | | | 199 | | | | | 120 | | |
| 53 Vienna (U) | | | 128 | 115 | 114 | | | 80 | 78 | 80 |
| 54 Sofia (U) | | | 196 | | | | | 208 | | |
| 55 Saint-Petersburg (U) | | | | 161 | | | | | 89 | |

**Table A3.** The cleaning cost savings for modern glass surfaces exposed outdoors in sheltered position due to 50% air pollution reduction (% of one cleaning investment, per year during the cleaning interval = %/year) (Equations (2), (3) and (5)), for the reported stations and years. The averages reported in Figure 4, were calculated as the mean of the averages over the stations for each year in the table, for Equations (2) and (3). The table represents the data availability for the ICP-materials stations and years. The input values for the environmental parameters to the calculations, for year 2002 until 2011, are reported in Reference [28], and for 2014 reported in Reference [36]. (U) = Urban, (I) = Industrial, (R) = Rural.

| | Cleaning Cost Saving Due to 50% Air Pollution Reduction (%/year) | | | | | | | | | |
| | Equation (2) | | | | | Equation (3) | | | | |
| Station Year: | 2002 | 2005 | 2008 | 2011 | 2014 | 2002 | 2005 | 2008 | 2011 | 2014 |
|---|---|---|---|---|---|---|---|---|---|---|
| 1 Prague (U) | | 176 | 143 | 168 | 154 | | 74 | 73 | 74 | 72 |
| 3 Kopisty (I) | | 138 | 146 | 174 | 177 | | 74 | 74 | 79 | 76 |
| 7 Waldhof-Langenbrügge (R) | 63 | | | | 53 | | | | | |
| 10 Bottrop (I) | 182 | 177 | 168 | 146 | 137 | 77 | 78 | 73 | 73 | 73 |
| 13 Rome (U) | | | | | 131 | | | | | 75 |
| 14 Casaccia (R) | | | | | 0 | | | | | 0 |
| 15 Milan (U) | | | | | 138 | | | | | 76 |
| 23 Birkenes (R) | | 0 | 0 | 0 | 0 | | 0 | 0 | 0 | 0 |
| 24 Stockholm south (U) | | | | 23 | 24 | | | | 12 | 6 |
| 26 Aspvreten (R) | | | | 0 | 0 | | | | 0 | 0 |
| 31 Madrid (U) | | | 56 | 128 | 110 | | | 52 | 69 | 62 |
| 33 Toledo (R) | | | 0 | 0 | 0 | | | 0 | 0 | 0 |
| 35 Lahemaa (R) | | 0 | 0 | | 0 | | 6 | 0 | | 0 |
| 40 Paris (U) | | 163 | 152 | | | | 74 | 73 | | |
| 41 Berlin (U) | 76 | | 127 | 159 | 152 | 82 | | 76 | 73 | 73 |
| 44 Svanvik (R) | | | | | 0 | | | | | 0 |
| 45 Chaumont (R) | 17 | | 0 | 0 | 0 | 12 | | 0 | 0 | 0 |
| 50 Katowice (I) | | 71 | 162 | 89 | 152 | | 94 | 78 | 87 | 79 |
| 51 Athens (U) | | | 86 | 122 | 107 | | | 85 | 76 | 79 |
| 52 Riga (U) | | | 140 | | | | | 73 | | |
| 53 Vienna (U) | | | 128 | 115 | 114 | | | 78 | 78 | 78 |
| 54 Sofia (U) | | | −1 | | | | | 91 | | |
| 55 Saint-Petersburg (U) | | | | 137 | | | | | 74 | |

## Appendix B

This appendix contains figures with separate results for modern glass surfaces exposed outdoors in sheltered position, calculated by Equations (2) and (3). They correspond to the averages of Equations (2) and (3) shown in Figures 4–6 in the main text.

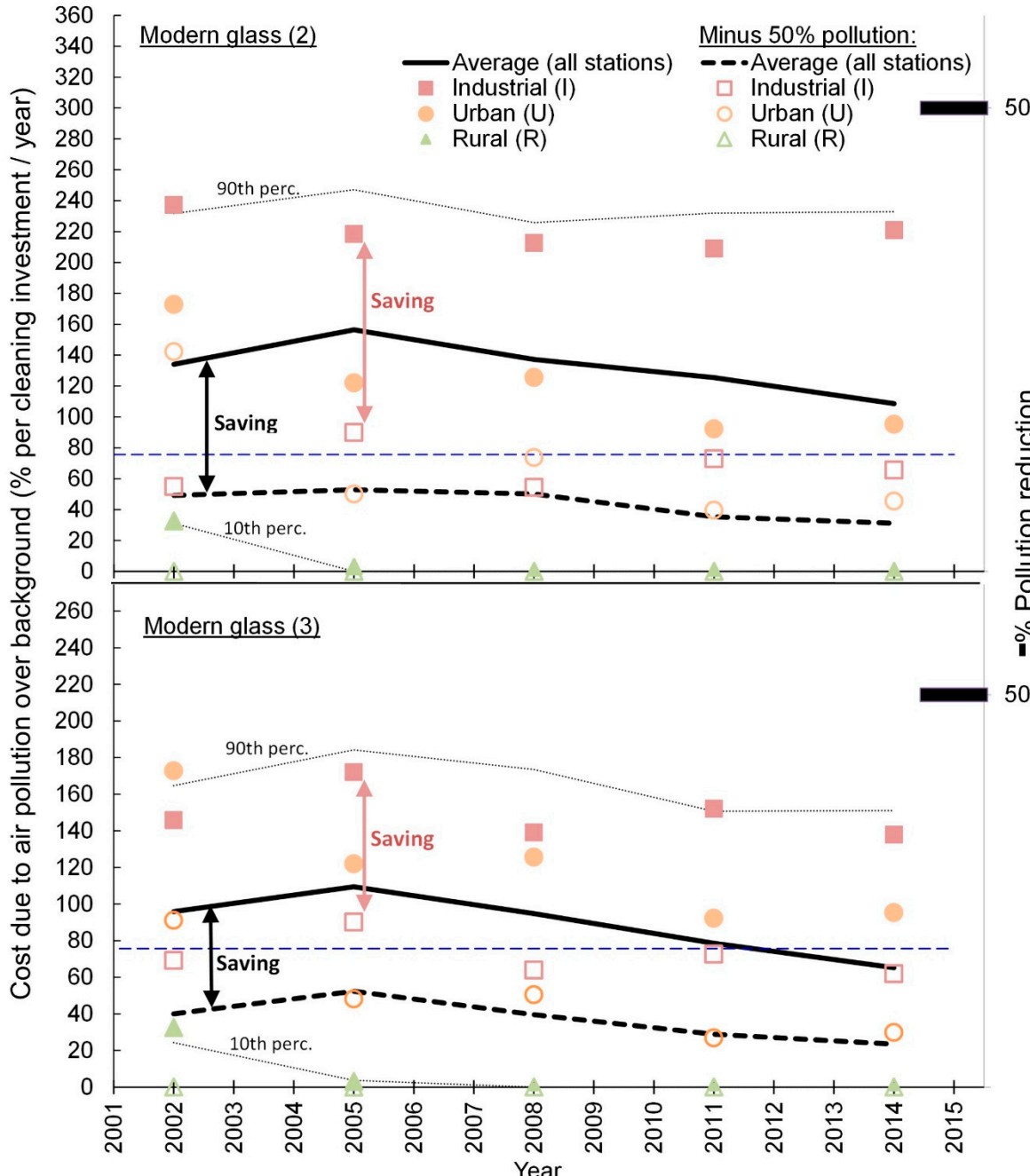

**Figure A1.** Cleaning costs and potential cost savings due to reduction in air pollution, calculated from Equations (2) and (3). The calculated average, and 10th and 90th percentile, for the cleaning costs due to air pollution at the measured values and due to a 50% reduction of the impacting pollutants (Equations (2)–(5)), compared with the tolerable level for modern glass surfaces exposed outdoors in sheltered position according to Equations (2) and (3) on all the ICP-materials stations with available data, and for the sub-selections of stations noted as urban (U), industrial (I) and rural (R), calculated for every year for all the stations with available environmental data (Appendix A).

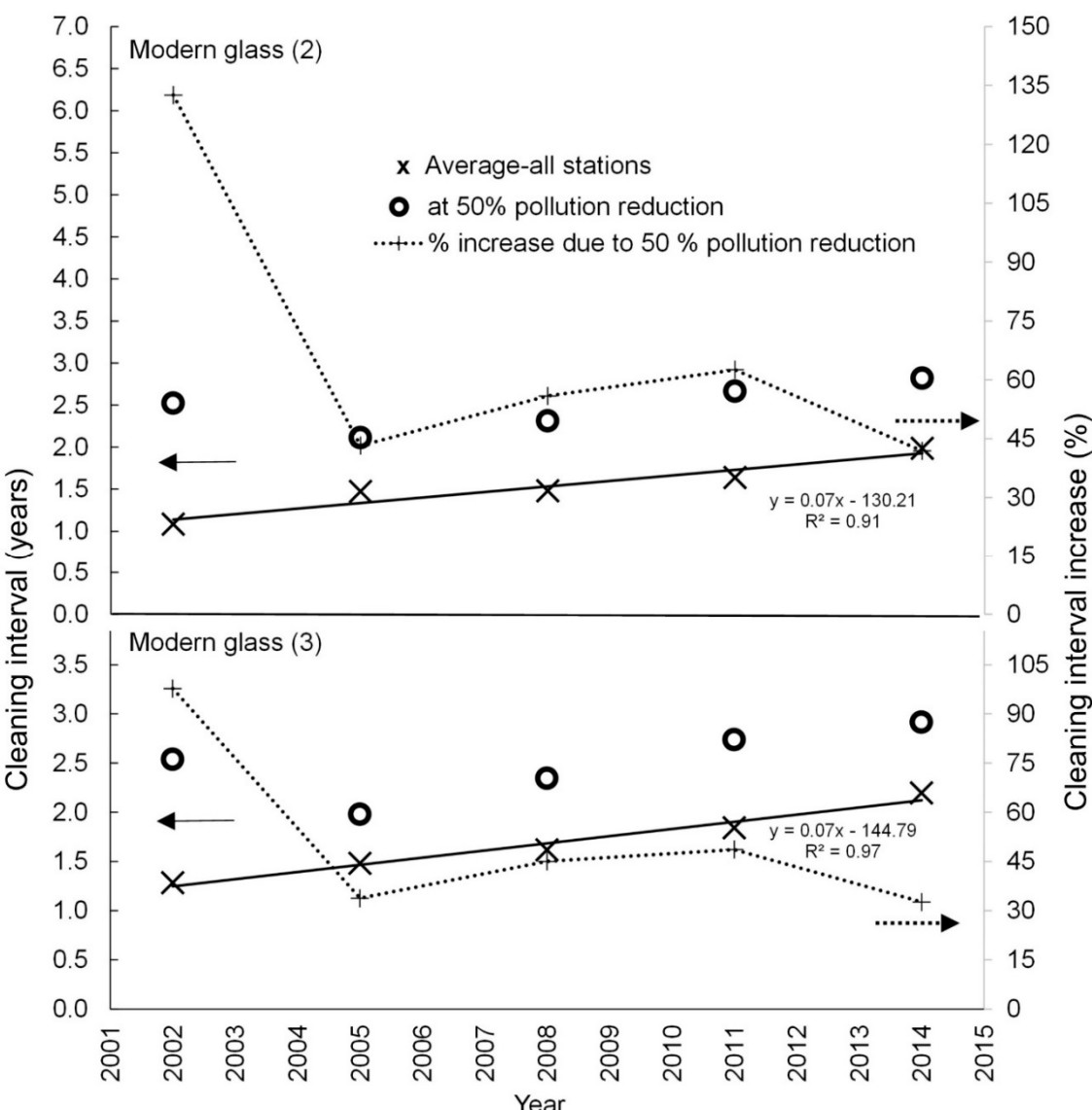

**Figure A2.** Cleaning intervals calculated from Equations (2) and (3). Indicative average intervals before recommended cleaning (years) and increase in cleaning intervals (%) of modern glass surfaces exposed outdoors in sheltered position due to 50% simultaneous reduction in the impacting pollutants according to Equations (2) and (3) for the ICP-materials stations since 2002. The calculations used a tolerable haze of 3% before cleaning, representing the average of the annual haze for all the ICP-materials measurement year and stations. The "% cleaning interval increase" is the difference between the cleaning interval at 50% pollution reduction and the average values for all the stations, given in the figure.

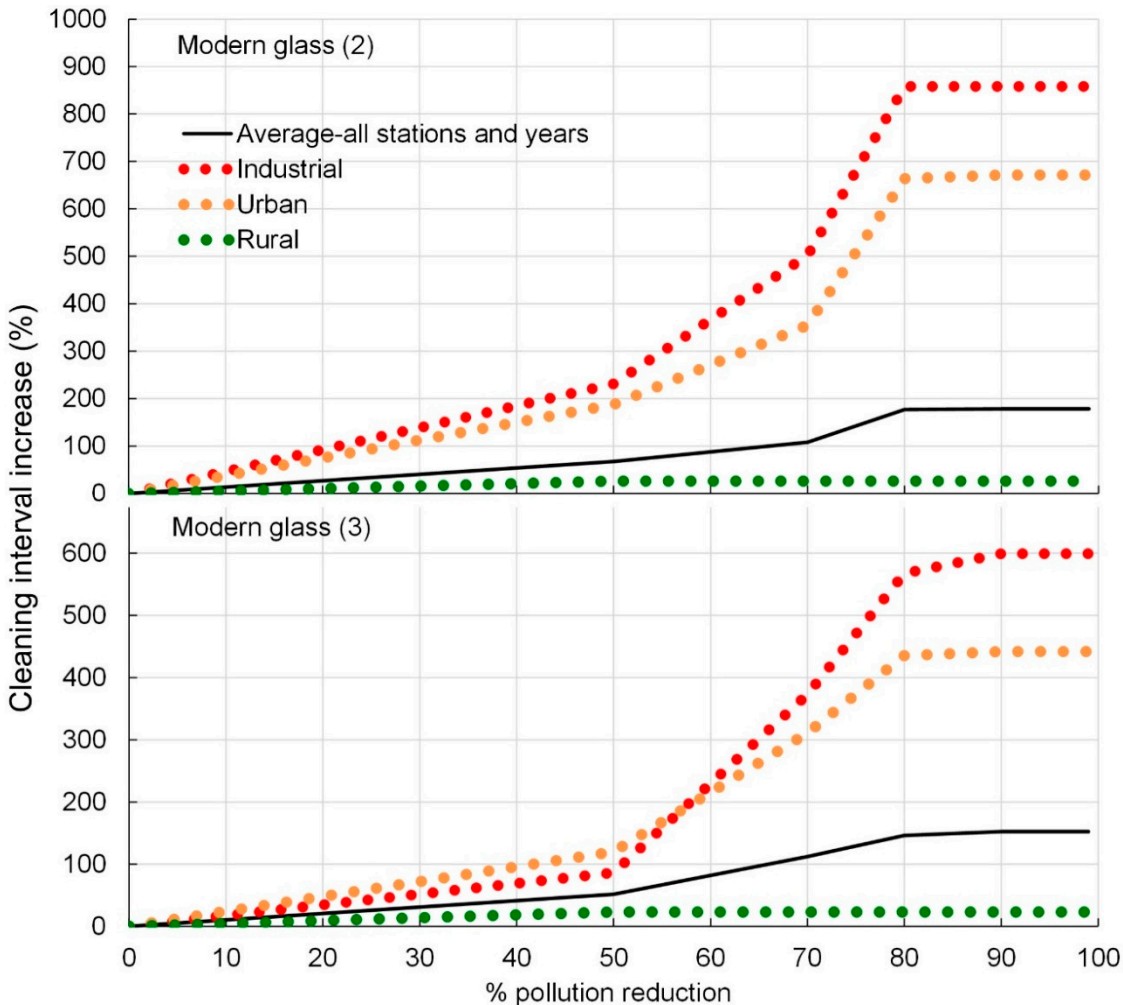

**Figure A3.** Cleaning interval increases due to hypothetical reduction in air pollution, calculated from Equations (2) and (3). Calculated average per cent increase in cleaning interval for modern glass surfaces exposed outdoors in sheltered position according to Equations (2) and (3), from 2002 to 2014, at the locations of the ICP-materials stations, and for the selections of industrial, urban and rural stations, due to increasing simultaneous percentages reduction of all the impacting pollutants. The calculations used a tolerable haze of 3% before cleaning, representing the average of the annual haze for all the ICP-materials measurement year and stations.

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
