# Peer review of "Cleaning Costs for European Sheltered White Painted Steel and Modern Glass Surfaces Due to Air Pollution Since the Year 2000"

_atmosphere, doi:10.3390/atmos10040167_

Round 1

Reviewer 1 Report

This manuscript reported the savings of the cleaning cost of two specific types of materials, the white painted steel surface, and the modern glass, due to the reduction of air pollution, based on data from the ICP-material project, which is a multi-year project across Europe. The topic of the manuscript is related to the focus of the special issue. Results are well presented and discussed. I would recommend the publication of this manuscript with some revisions.

Major comments:

1. The current introduction part lacks a continuous path that takes the reader from the big picture to the specific goal of this work. The description of this work scattered in different parts of the introduction. The author should summarize the goals of the manuscript at the end of the introduction.

2. The significance of this work could be better emphasized in the introduction and the conclusion part. Is the reduction of the building cleaning cost going to offer a major help for some local government agencies? Is the method presented in this manuscript going to help assess the air quality damage to buildings in other parts of the world?

3. The assumptions of equations are presented in the draft, but there’s a lack of description that how the conditions of your analysis ties to these assumptions. For example, in line 148-149, does your analysis falls inside these time and air quality ranges?

4. The air quality data used in this work should be better described or presented. Table 1 only presented some mean values in 2002, 2005, and 2014. Were the SO2, NO2, and PM10 continuously decreasing from 2002 to 2014? Was the air quality data being measured continuously and recorded on a daily basis? How’s the variation of the air quality data?

5. Did the air quality data you measured from 2002-2014 show a half reduction? If not, how would you justify that half reduction of air pollution is a representative case?

Some detailed comments:

Line 134 – How was the PM10 range in that one year? Is it similar to your analysis?

Line 174 – Are you assuming the pollution reductions in each year are always the same?

Line 175-178 – I am confused. Probably need a better explanation here.

Figure 3 – It could be useful to note down the type of sites, urban, industrial, or rural. Maybe add some country names to the map.

Line 259 – It would be helpful to give some explanation here that why to compare background with WHO guideline and EU AQD.

Line 266 – 285 – Is it possible to present these comparisons in a table? This paragraph is hard to follow.

Line 294 – 303 – Did you calculate the cleaning cost here using equation 4? If so, it seems like if it is for the same building material, even in different locations, the cleaning cost due to air pollution should be directly comparable, is it right?

Figure 7 – It would be helpful to get rid of the contour lines, leave the color here, and put the city map as the base map. 

Author Response

1 and 2. We have reorganized the Introduction to obtain a better continuous path in the description.

We have moved some sentences about the goal to the end of the Introduction, and reformulated the last three paragraphs to better summarize the goals and significance of the manuscript, also pointing to themes in the Discussion.  We also in these paragraphs explain limits of this study, in relation to other deterioration impacts, mechanisms and costs, and in relation to application of the method outside of the range of the input data, for example outside Europe. A final sentence was also added in the Conclusion, putting the soiling and costs into the total perspective of environmental loads and maintenance costs for buildings.

3. Regarding Eq, 1: This was already partly addressed, among other places in the Discussion chapter by the following sentence:

“The development of the soiling function for the white painted steel (1) was based on one year of experimental data. It is uncertain how well the first year development of the soiling describes the further progress of the soiling over many years [10].”

Regarding Eq, 2-3: The following sentence was added (l. 163-165):

“In the present study the maximum time was set to 1500 days, equal to the cleaning interval in the background (see section 2.4) and the concentration criteria were fulfilled (see Data section).”

4. The data and trends were described in more detail (l. 213-223)

5. The argument to show results for a hypothetical 50% reduction was given in the second last paragraph of the Introduction. This hypothetical reduction is not related to any actual reduction trend. The word “hypothetical” was now added to this introductory explanation to be clear about this, as it was used later throughout. The costs (and changes/savings) due to the measured changes in pollutants, over the years, are reported directly as the main and first results in Figure 4 and Table 3. Then the results of the hypothetical reduction of 50% are reported (Figure 4 – change to dashed line and Table 4). (Later figures develop the theme of actual costs (full line) and hypothetical savings (dashed line in Fig 5 and Fig. 6). ..“hypothetically” was also added in the Figure 7 text.

Detailed comments:

1.       The PM10 range for the experiments by Watt et al., was now included and a comparison and evaluation of application with ICP-materials data was made (line 140-145).

2.       Yes in this simulation of the hypothetical 50% reduction, this reduction is assumed for every station and future year.

In the methods section 2.1 (l. 173-175) this was already explained as follows, which should cover this:

“The present average expected cleaning frequency was calculated for the ICP-materials locations, for each year of the environmental measurements, assuming constant future environments as in the measurement years. The cleaning frequencies for the hypothetical 50% reduced pollution situations were calculated similarly.”

3.       The following sentence was added at the end of section 2.2 (line 193-196)

“Thus, by the terms in Eq. (5), the reporting of results in this work is as 100 ∙ (Kp – Kr) / ΔC (% of one cleaning investment, per year), rather than 100 ∙ (Kp – Kr) / Kp (% of the yearly cleaning cost).”

4.       Figure 3: The station type was added to the figure and the following text was added to the caption: “I = industrial, U = urban, R = Rural. Station names and data points are given in Table A1 to A3.”

5.       A footnote “6” was added to supply some more information why “comparison with health related guidelines and directives, which mostly guide policy” are important to this work.

6.       We understand that this explanation of the derivation of the cleaning intervals for the guidelines, is a little tedious in its detail. It is needed background information to the cost estimations and need to be included. We don’t want to give this more emphasis by producing tables etc. for it. Also, as explanations/arguments are needed in between in this derivation putting it into table may not help much. This paragraph could possibly have been moved to the Appendix. However, we prefer to keep it with the text in this section (2.4) where it explains the Table 2 results. We suggest instead that this paragraph can be indented in full, given an improved introduction sentence and a slight gray tint (if possible?) to indicate its nature. (now done)

7.       Yes. It is right. Al the calculations are for the white painted steel and modern glass, according to Eq. (4), as described before, in Section 2.2.

8.       This map could surely have been made differently. The production of the map involves much work, and we now need to keep this format (except possible simple editing of text etc. which could of course be possible)

Reviewer 2 Report

The manuscript deals with the effect of air pollution on cleaning costs of materials representing transparent (glass) and opaque (painted steel) surfaces, estimating savings due to experienced and potential future air quality improvement. There are still few studies on economy of pollution and data obtained in the ICP Materials programme are highly valuable in this sense. The paper is very well written and logically organized. The conclusions are supported by solid data and all major limitations of the study are identified and discussed. Therefore, I recommend the paper for publication in Atmosphere.

Several minor remarks to be considered by the authors follow.

-          I miss some data on the materials used in the study. At least a basic characterization of the glass (composition, surface roughness, …) and white painted steel (shade, chemical nature of topcoat paint, roughness, Tg, …) should be given. It would also help to discuss how much eventual material variations might affect the results. E.g., the surface roughness, composition, hue. Even an estimate would be useful.

-          Line 130: The constant unit should be m3/μg.day.

-          Figure 4: The authors are trying to show too many data at once. The charts are then difficult to read. It might be better to skip the data for site categories as they are present in Table 4 anyway.

-          Figure 6: Consider reducing the x-axis range to e.g. 80, or even 50 %. Like this, the effect of more realistic, lower pollution reductions is impossible to see.

-          Table 5: Only line 2 appears correct to me. The absolute cost in Euro/m2.year should be equal to the empirical cost in Euro/m2 divided by the maintenance interval in years, according to my understanding of the paper. This is not true for lines 1, 3 and 4. Please, verify the data correctness.

-          Line 650-653: I am not sure these data are correct either. In Table 5, the value of 65 %/year is given for glass but values of 95 and 65 %/year are given in the text. Similarly, the ranges and values for the other materials are different in Table 5 and in the text.

Author Response

1.       We add at the start of the methods chapter, where the materials are described that the glass was “modern silica-soda-lime glass (Si-Na-Ca float glass, Planilux®) [24]” and also insert the most relevant reference for the white painted steel: [31]. The references to the experiments with the soiling of the white painted steel, do not describe the paint or metal in more detail (as far as we can see). For more information about the glass, the reader should consult the reference(s) (now inserted). Both the modern glass and the white painted steel could probably be considered “standard materials”. The developers of the DRF for the white painted steel may have considered that detailed characterization of the steel and paint was of little importance for the soiling (?).  We do not, however, want to speculate in this.

 We agree about the interest in the importance of materials variation for the soiling. This was discussed in the Discussion chapter in relation both to the physical soiling and the cleaning costs (for example lines 587-592 and 710-715). Considering the complexity of this topic we are not aware of a simple estimate of, for example surface roughness (choosing one surface property) – which can be related to soiling on documented surfaces and the respective cleaning costs. It was beyond the scope of this paper to do experimental tests, or discuss possible estimates of this: Considering the large variation and differences in building and monument surfaces this is a huge topic, which could probably be a separate theoretical / discussion / review paper. New well-designed experimental work could probably increase knowledge about this.

2.       The unit was corrected

3.       Figure 4: Table 4 shows the hypothetical cost savings (due to 50% pollution reduction). These are the differences (illustrated by the arrows) in Figure 4, which shows directly the cost over background based on the measured pollution. These first hand cost data are not shown in Table 4/elsewhere. The values for the sub-sets give added information about variation.  We agree that the figure was somewhat confusing to read and have toned down the markers for the sub-sets and toned up the main average cost result. – The same was done with Figure B1.

4.       Figure 6: We see the point of the reviewer. We changed the range on the x-axis to be similar: 0-800 %.

5.       Table 5: We understand the comment of the reviewer. For any single station the reviewer is correct, - in the circumstance when the background cost is not subtracted. As is stated in the table text (and throughout the paper) the reporting here is of costs over background. However, this has the opposite effect on the value (contrary to the discrepancy noted by the reviewer) – it gives lower absolute reported cost. The reason for the longer maintenance intervals than expected is that these are reported averages for many stations, which give a higher weight to long time - maintenance intervals. This was already explained in the results section (line 483-491). However this is much more apparent when reading Table 5 and we have added an explanation just before the table:

“As was noted in Section 3.3 it should be stressed again that the average absolute yearly costs over background for all the stations, reported in Table 5, are not simple divisions of the total empirical costs by the maintenance interval. This would be the case for the absolute yearly cost for any single station when including the background cost, but not so for the averages of all the stations. In Table 5 it is coincidentally so for the white painted steel. If, however, including the background cost the average stations absolute cost for the cleaning of the white painted steel would be 5.4 Euro/m2·year.”

A sentence reporting the calculated background costs (%/year) was added in the start of Section 3.1 (l. 370-374)

6.       Line 703-706 (previous 650-653): The text is now written:

Table 5 shows differences in possible relative cost, and cost savings from reduction in air pollution (with 50%). They are more than one order of magnitude higher for the cleaning of sheltered glass (cost of 95 %/year and cost savings of 65 %/year) than for the cleaning of white painted steel surfaces (cost of 4.0 and cost savings of 3.5 %/year).

This cost and cost saving (95 and 65 %/year) are both given in Table 5 (row 1, column 3 and 5), then they are compared with values below in the same columns for different materials. Mistakenly, (and confusingly) “limestone” was written instead of “steel” (line 705). This was corrected. Through this paragraph (lines 703-709), the words cost and cost savings were now added before all numerical values to the make the reading easier.